# 📝 EviNote-RAG: Enhancing RAG Models via Answer-Supportive Evidence Notes

## Abstract

Retrieval-Augmented Generation (RAG) has advanced open-domain question answering by incorporating external information into model reasoning. However, effectively leveraging external information to enhance reasoning presents the following challenges: (1) *low signal-to-noise ratio*, where answer-supportive external information is diluted by irrelevant material, and (2) *error accumulation*, which arises in multi-hop reasoning when incomplete or misleading information is incorporated. To address these challenges, we introduce **EviNote-RAG**, a framework that follows a retrieve–note–answer workflow. Instead of reasoning directly over raw external information, the model first produces *Supportive-Evidence Notes (SENs)*, which concisely preserve answer-critical information and explicitly mark key and uncertainty information to improve accuracy. We further design an entailment-based *Evidence Quality Reward (EQR)* to ensure that SENs are logically sufficient to derive the final answer, thereby enhancing SENs' quality. Experiments on both in-domain and out-of-domain QA benchmarks show that EviNote-RAG achieves state-of-the-art performance, improving answer accuracy, training stability, robustness, and efficiency. In particular, it yields relative F1 gains of **20%** on HotpotQA (+0.093), **40%** on Bamboogle (+0.151), and **91%** on 2Wiki (+0.256), benefiting from improvements in the reasoning process.

## 1 Introduction

Large Language Models (LLMs) have evolved from next-token predictors into systems capable of advanced reasoning (Chowdhery et al., 2022; Verma et al., 2024b; Zheng et al., 2025; Yin et al., 2025; Jiang et al., 2025). Since the factual knowledge of LLMs is fixed during pre-training, they are prone to generating incorrect or outdated information when deployed in real-world tasks where knowledge evolves rapidly (Ji et al., 2022; Zhang et al., 2025). To address this limitation, Retrieval-Augmented Generation (RAG) (Arslan et al., 2024; Gao et al., 2025) has emerged by incorporating a search tool that supplies up-to-date external evidence at inference time, enabling models to ground their responses in timely information and improve factual consistency.

Despite recent advances, how to effectively leverage external documents to support reasoning remains a fundamental challenge (Gao et al., 2025). Prompt-based methods address this through multi-step reasoning (Jiang et al., 2024; Tran et al., 2024; Xiong et al., 2025) or adaptive workflows (Lee et al., 2024; Zhou et al., 2024; Wu et al., 2025a; Li et al., 2025a; Zhao et al., 2025b), while tuning-based strategies (Liu et al., 2024; Zhang et al., 2024a) improve fine-grained information extraction but often sacrifice generalization. More recently, advances in Reinforcement Learning (RL) (Kaelbling et al., 1996; Guo et al., 2025) have inspired RL-based RAG approaches (Zhang et al., 2024a; Wei et al., 2025a; Song et al., 2025b; Jin et al., 2025; Li et al., 2025a; Deng et al., 2025), which surpass earlier paradigms by exploring optimal strategies and enhancing generalization. Yet, RL-based RAG methods still rely on outcome-based rewards that evaluate only final correctness, offering little guidance for intermediate reasoning. Consequently, models remain constrained to the *retrieve-then-answer* paradigm, facing two persistent obstacles: (1) **Low Signal-to-Noise Ratio (SNR)**, where retrieved evidence often includes substantial irrelevant content, making supportive information sparse (Shi et al., 2023; Jin et al., 2024); and (2) **Error Accumulation**, where reasoning errors (Shi et al., 2023) amplify when inference depends on incomplete or noisy evidence, especially

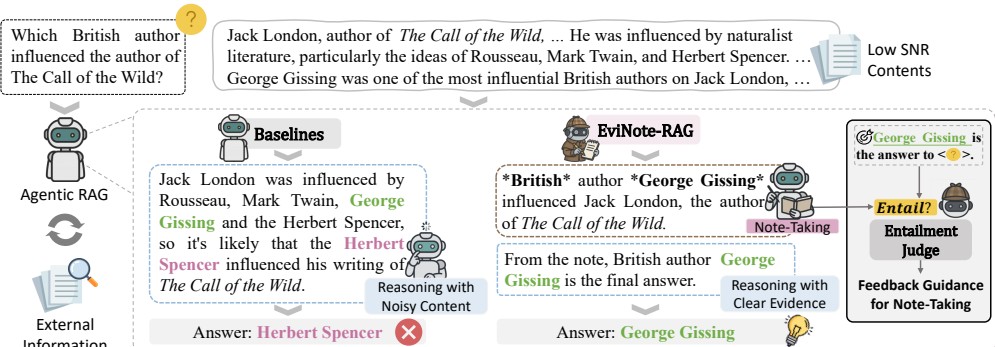

Figure 1: EviNote-RAG vs. Baselines (Song et al., 2025b; Jin et al., 2025): EviNote-RAG distills key information through evidence notes and, guided by an Entailment Judge, ensures that retained content directly supports the answer, thereby mitigating noise and enhancing performance.

in multi-hop QA. Addressing these issues calls for RL strategies that not only boost performance but also equip models with more effective workflows for handling long and noisy contexts.

To address the core limitation of existing RAG systems, we propose **EviNote-RAG**, an end-to-end RL-based RAG framework that restructures the pipeline into a *retrieve–note–answer* process (Fig. 1). EviNote-RAG trains LLMs to generate *Supportive-Evidence Notes (SENs)*, concise abstractions that preserve only answer-critical information and discard irrelevant content to improve answer accuracy. Each SEN further highlights *key* and *uncertain* information, echoing human note-taking strategies to improve focus and reduce misleading reasoning. Most importantly, we formulate evidence selection as a reinforcement learning problem: through the *Evidence Quality Reward (EQR)*, a lightweight entailment judge evaluates how well each SEN supports the final answer. This reward signal encourages the model to explore strategies for evidence extraction while being guided toward more accurate and faithful use of information. As a result, EviNote-RAG achieves end-to-end optimization that reduces noise, mitigates error accumulation, and enables the model to learn an effective strategy for precise information utilization.

We validate our approach through extensive experiments on both in-domain and out-of-domain QA benchmarks, and summarize our main contributions as follows:

- We propose **EviNote-RAG**, a structured agentic RAG framework that transforms the standard retrieve-then-answer paradigm into a retrieve–note–answer pipeline, improving content distillation and reasoning reliability.

- We introduce a human-inspired *Retrieval-based Summarization* mechanism that generates Supportive-Evidence Notes (SENs), highlighting key and uncertain information to enhance focus and mitigate noise in retrieved content.

- Our approach not only achieves state-of-the-art performance across multiple QA benchmarks, but also significantly improves training robustness. For example, relative to the Base model, EviNote-RAG lifts F1 by 20% on in-domain HotpotQA (+0.093), 40% on OOD Bamboogle (+0.151), and 91% on 2Wiki (+0.256). Moreover, denser, better-shaped reward signals and reduced verbosity yield more stable, sample-efficient training.

## 2 RELATED WORK

### 2.1 INSTRUCTION-GUIDED RAG METHODS

Instruction-guided methods (Amplayo et al., 2022; Yao et al., 2023; Jeong et al., 2024; Jiang et al., 2024; Wu et al., 2025a) enhance RAG by designing prompts that automate retrieval and guide multi-step reasoning (Jiang et al., 2024; Xiong et al., 2025). These approaches (Zhang et al., 2024b; Li et al., 2024a) typically decompose questions into sub-problems, retrieve external knowledge, and synthesize structured answers (Zhou et al., 2024; Zhao et al., 2025b; Tran et al., 2024). Other works (Li et al., 2025a; Lee et al., 2025) integrate retrieval directly into the reasoning loop, while

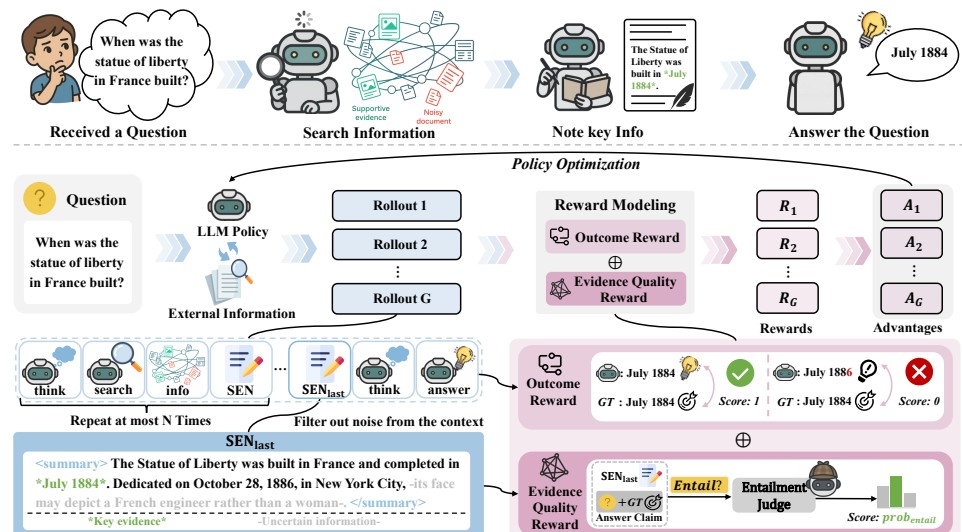

Figure 2: Overview of the EviNote-RAG. To improve information utilization, the method introduces a note-taking phase where the model generates Supportive-Evidence Notes (SENs) that capture only the information necessary for answering. An entailment-based Evidence Quality Reward (EQR) further ensures that each note faithfully supports the final answer, guiding the model toward more accurate and evidence-grounded reasoning.

more recent efforts (Yue et al., 2024; Verma et al., 2024a; Li et al., 2025a; Alzubi et al., 2025; Feng et al., 2025) interleave retrieval and reasoning adaptively. Despite these advances, prompt-based approaches inherently depend on the foundation model's generalization ability in RAG, which remains limited. Our framework instead employs a post-training, reward-driven objective that explicitly shapes information selection and reasoning, leading to more faithful and task-adapted performance.

## 2.2 REWARD-GUIDED AGENTIC RAG

Reward-guided approaches (Zhang et al., 2024a; Guan et al., 2025; Huang et al., 2025; Zhao et al., 2025a; Wang et al., 2024b; Wei et al., 2025a; Deng et al., 2025) employ Reinforcement Learning (RL) (Kaelbling et al., 1996) to optimize reasoning policies through scalar feedback derived from task performance. Early work (Nakano et al., 2021) demonstrated that reward signals can effectively guide multi-step retrieval and improve factual accuracy. Building on recent advances in RL (Guo et al., 2025), RL-based RAG approaches (Qi et al., 2025; Chen et al., 2025; Jin et al., 2025; Wei et al., 2025b) have surpassed previous paradigms by enabling optimal strategy exploration and improved generalization. Subsequent works have broadened this framework to diverse scenarios and tool use (Zheng et al., 2025; Dai et al., 2025; Wang et al., 2025; Sun et al., 2025a; Gutiérrez et al., 2025; Shao et al., 2025), highlighting the performance gains from RL-based supervision. However, most existing reward-guided methods (Wu et al., 2025b; Song et al., 2025b; Sun et al., 2025b) operate directly on raw, often noisy passages, which leads to a low signal-to-noise ratio (Shi et al., 2023; Jin et al., 2024) and error accumulation across multi-hop reasoning. EviNote-RAG tackles this limitation by using supportive-evidence notes to structure retrieved information and by applying supervision that enforces logical consistency between the notes and the final answers. These mechanisms together promote more reliable reasoning and improve answer accuracy.

## 3 METHODOLOGY

This section presents EviNote-RAG (as shown in Fig. 2), which integrates Supportive Evidence Notes (SENs) to distill answer-relevant content from retrievals and an Evidence Quality Reward (EQR) to ensure each note faithfully supports the final answer. Together, these components guide more accurate and robust reasoning. The following subsections describe the pipeline, SEN design, and reward formulation.

## 3.1 EVINOTE-RAG PIPELINE

Upon receiving a query, **EviNote-RAG** either issues `<search>` to retrieve external evidence or, when sufficiently confident, answers directly. Retrieved content arrives as `<information>` and may be noisy; therefore, the system produces *Supportive-Evidence Notes (SENs)* in `<summary>` to filter distractors and retain evidence critical to the answer. Once sufficient notes are consolidated, the agent finalizes the response in `<answer>`. Importantly, SENs explicitly link supportive evidence to the evolving answer, ensuring consistency and precision. The following subsections provide further details.

## 3.2 SUPPORTIVE-EVIDENCE NOTE

To improve information utilization, our method uses Supportive-Evidence Notes (SENs) within `<summary>` tags to filter out irrelevant content, ensuring retention of supportive evidence. Next, we detail two key components of SENs: evidence-aware annotations and the dynamic SEN workflow, which jointly enhance information utilization and strengthen model performance.

**Evidence-aware Annotations.** To enhance information utilization, SENs incorporate two annotation types: *key information* (denoted by * ) and *uncertain information* (denoted by – ). These annotations preserve the model's certainty in multi-turn interactions, enabling precise identification of key information and avoiding misguidance from uncertain data, yielding substantial gains over naive summarization (Section. 4.6, Naive Summary vs. SEN).

**Dynamic SEN Workflow.** We emphasize that SEN generation is optional after each retrieval phase, allowing the model to dynamically determine the necessity of summarization based on retrieval outcomes. This dynamic workflow design is essential for enhancing RL training effectiveness (Section. 4.6, Force Summary vs. Ours), underscoring the importance of flexibility in achieving optimal RAG strategies. Furthermore, To guide high-quality SEN generation, we propose the Evidence Quality Reward (EQR), which provides entailment-based feedback to focus on answer-relevant content. Details are presented in the next section.

## 3.3 EVIDENCE QUALITY REWARD

To encourage the generation of high-quality SEN, the Evidence Quality Reward (EQR) introduces an Entailment Judge as a source of supervision. The underlying intuition is straightforward: a well-formed SEN should provide sufficient grounds for logically inferring the ground-truth answer. We realize the Entailment Judge through a lightweight Natural Language Inference (NLI) model (e.g., DistilBERT (Sanh et al., 2019)), which evaluates whether the final SEN entails the correct answer. To be more specific, we first construct an answer claim $h$ that asserts the ground-truth answer $\text{ANS}_{\text{gt}}$ is the correct answer to the question $q$. We then use the final SEN $\text{SEN}_{\text{last}}$ as the input text, and evaluate whether it logically supports $h$ using the Entailment Judge model $\mathcal{M}_{\text{Judge}}$, as shown below:

$$R_{\text{EQR}} = \mathcal{M}_{\text{Judge}}(\text{SEN}_{\text{last}}, h)[\text{entailment}], \tag{1}$$

where $[\text{entailment}]$ denotes the confidence score assigned to the entailment class. This reward $R_{\text{EQR}} \in \mathbb{R}$ encourages the model to generate SENs that logically support the correct answer. To reduce computational overhead, EQR is applied only to the final SEN in each output sequence. For example, given the question *"What is **the largest** planet in the solar system?"* with the ground-truth answer *"Jupiter"*, we construct the answer claim *"Jupiter is the answer to 'What is the largest planet in the solar system?'"*. If the SEN states *"Jupiter is the largest planet in the solar system"*, the entailment score is high. In contrast, if it only states *"Jupiter is a planet in the solar system"* while omitting the crucial fact of being *the largest*, the score is low. This example demonstrates how subtle semantic distinctions in SENs affect whether the answer can be logically inferred, underscoring the importance of entailment-aware generation.

## 3.4 TRAINING STRATEGY

**Reward Strategy.** We design a reward strategy to supervise the model's behavior throughout training. This strategy balances two goals: (1) encouraging the model to explicitly mark uncertainty and

highlight key information when answer prediction is unreliable; (2) promoting accurate and well-supported answers as performance improves. The scalar reward $R(\cdot)$ is computed as:

$$R = \begin{cases} 1 + R_{\mathrm{EQR}} & \text{format } \checkmark, \text{ answer } \checkmark \\ 0.1 + R_{\mathrm{EQR}} & \text{format } \checkmark, \text{ answer } \times, \text{ note-taking } \checkmark \\ 0 & \text{otherwise} \end{cases} . \tag{2}$$

Here, $\checkmark$ indicates satisfaction and $\times$ a violation. The **format** criterion holds if the output includes an explicit `<answer>` tag, at least one `<summary>` tag, and follows the prescribed schema. The **answer** criterion requires an exact match with the ground truth, and the **note-taking** criterion holds when Evidence-aware Annotations are marked according to the SEN design. This reward design ensures that the model is gently guided to perform structured note-taking when its QA capability is still developing (through a small reward of $0.1 + R_{\mathrm{EQR}}$), and increasingly incentivized to generate precise, logically supported answers when it becomes more reliable (through $1 + R_{\mathrm{EQR}}$). Note that this strategy integrates the Evidence Quality Reward (EQR) $R_{\mathrm{EQR}}$, which provides entailment-based feedback to further emphasize relevance and faithfulness in the final generated answers.

**Policy Optimization.** In this work, we adopt the GRPO algorithm (Shao et al., 2024) to optimize the policy $\pi_\theta$ using the reward $R$. GRPO updates the current policy $\pi_\theta$ using a reference policy $\pi_{\theta_{\mathrm{ref}}}$ and a set of rollouts generated by a previous policy $\pi_{\theta_{\mathrm{old}}}$. The training objective is extended and formulated as follows:

$$r_1, r_2, ..., r_G = R(y_1, y_2, ..., y_G), \tag{3}$$

$$A_i = \frac{r_i - mean(r_1, r_2, ..., r_G)}{std(r_1, r_2, ..., r_G)}, \tag{4}$$

$$\mathcal{J}(\theta) = \mathbb{E}_{x \sim \mathcal{D}, \{y_i\}_{i=1}^G \sim \pi_{\theta_{\mathrm{old}}}(\cdot|x)} \left[ \frac{1}{G} \sum_{i=1}^{G} \min \left( \frac{\pi_\theta(y_i|x)}{\pi_{\theta_{\mathrm{old}}}(y_i|x)} A_i, \right. \right.$$

$$\left. \mathrm{clip} \left( \frac{\pi_\theta(y_i|x)}{\pi_{\theta_{\mathrm{old}}}(y_i|x)}, 1 - \epsilon, 1 + \epsilon \right) A_i \right) \tag{5}$$

$$\left. - \beta \, \mathbb{D}_{\mathrm{KL}} \left( \pi_\theta \, \| \, \pi_{\theta_{\mathrm{ref}}} \right) \right]$$

where, $x$ denotes an input sampled from the experience distribution $\mathcal{D}$, $y_i$ denotes an trajectory generated by $\pi_\theta$, $r_i$ denotes the reward assigned to $y_i$; $A_i$ represents its corresponding advantage. $\mathbb{D}_{\mathrm{KL}}$ denotes the unbiased estimator of KL divergence (Shao et al., 2024), and $\epsilon$ and $\beta$ are hyperparameters for balancing exploration and exploitation.

## 4 EXPERIMENTS

### 4.1 DATASETS

We evaluated EviNote-RAG on seven widely used Question Answering (QA) benchmark datasets: **(1) In-Domain Datasets** consist of NQ (Kwiatkowski et al., 2019) and HotpotQA (Yang et al., 2018). These datasets are considered in-domain because they originate from the same question distribution, allowing for a direct comparison on familiar tasks. **(2) Out-of-Domain Datasets:** Out-of-Domain Datasets include PopQA (Mallen et al., 2022), TriviaQA (Joshi et al., 2017), 2WikiMulti-HopQA (2Wiki) (Ho et al., 2020), Musique (Trivedi et al., 2022) and Bamboogle (Press et al., 2022). These datasets involve more complex multi-hop reasoning and are classified as out-of-domain because their question distributions differ significantly from our fine-tuning set. For testing, we randomly select 500 samples from each of the datasets, except for Bamboogle, where we use the entire 125 samples from its validation set.

### 4.2 METRICS.

We evaluated the model using the following metrics: **(1) Exact Match (EM)** evaluates whether the predicted answer strictly matches the ground truth, while **(2) F1 Score** balances precision and recall,

offering a more flexible measure when answers are close but not identical to the ground truth. These two metrics complement each other, providing a more comprehensive assessment of the accuracy of the answer. Furthermore, to demonstrate the impact of improving the quality of SEN support in the Ablation Study section, we introduce **(3) Evidence Quality Reward (EQR):** a metric designed to assess the quality of supporting evidence nodes (SEN), focusing on their relevance and logical consistency.

### 4.3 BASELINES

To evaluate our model, we compare it against several established baselines: **(1) Foundational Model:** All of our models use the Qwen-2.5-7B-Instruct model (Yang et al., 2024) as the foundational model. This includes *Direct Inference*, which generates answers without using retrieved context, and RAG WORKFLOW, which guides the foundational model for RAG retrieval solely by modifying instructions, without any additional training or fine-tuning. **(2) Chain-of-Thought (CoT) Methods:** This category includes RECAT and IRCOT (Trivedi et al., 2023), which enhance reasoning by explicitly generating intermediate Chain-of-Thought reasoning steps. **(3) Prompt-Based Agentic RAG:** This group includes models such as SELF-ASK (Press et al., 2022), ITER-RETGEN (Shao et al., 2023), SELF-RAG (Asai et al., 2024), and SEARCH-O1 (Li et al., 2025b), which combine retrieval and reasoning through designed prompts. We also include CR-PLANNER (Li et al., 2024b), which employs Monte Carlo Tree Search (MCTS) for planning. **(4) RL-Based Agentic RAG:** This category includes models like REARTER (Sun et al., 2025b), R1-SEARCH (Song et al., 2025a), and SEARCH-R1 (Jin et al., 2025), which extend the traditional RAG paradigm by incorporating agentic search and policy learning through reinforcement learning, enabling the model to adaptively refine its retrieval and reasoning strategies during training.

### 4.4 IMPLEMENTATION DETAILS

Our experimental framework is built upon the Qwen-2.5-7B-Instruct model (Yang et al., 2024) using the Verl framework (Sheng et al., 2025). The retrieval module utilizes the 2018 Wikipedia dump (Karpukhin et al., 2020) as the knowledge corpus, with the E5 model (Wang et al., 2024a) serving as the dense retriever. For model optimization, we apply loss masking to update only the tokens generated by the model. The learning rate is set to 1e-5, with a sampling temperature of 1.0. Training is performed with a batch size of 600 (distributed across 15 NVIDIA A100 Tensor Core GPUs), generating 4 rollouts per sample and limiting the maximum retrieval count to 5. In addition, one separate GPU is used to run a 144M-parameter DISTILBERT (Sanh et al., 2019) model for calculating the Evidence Quality Reward.

### 4.5 MAIN RESULTS

The overall performance of EVINOTE-RAG is summarized in Tab. 1, which reports results across both in-domain and out-of-domain benchmarks.

**In-Domain Performance.** On benchmarks whose distributions are aligned with the training data, EVINOTE-RAG achieves strong results and consistently surpasses baseline models. On HotpotQA, a dataset that requires complex multi-hop reasoning, EVINOTE-RAG significantly outperforms both RAG and CHAIN-OF-THOUGHT (CoT) methods. These improvements arise from the Supportive-Evidence Notes (SEN) mechanism, which filters out spurious retrievals, and the Evidence Quality Reward (EQR), which encourages the selection of answer-critical evidence. Together, these mechanisms enable the model to construct faithful reasoning chains and maintain factual consistency, thereby yielding more accurate in-domain answers.

**Out-of-Domain Generalization.** Across out-of-domain benchmarks, EVINOTE-RAG achieves clear gains over the strong RL baseline (Search-R1): +91% F1 on 2Wiki (0.536 vs. 0.280, +0.256), +40% on Bamboogle (0.528 vs. 0.377, +0.151), +23% on Musique (0.336 vs. 0.274, +0.062), and +5.4% on TriviaQA (0.795 vs. 0.754, +0.040), with near-parity on PopQA (0.491 vs. 0.498). These gains are driven by behavior shaping: Supportive-Evidence Notes (SEN) compress question-conditioned evidence before generation, and the entailment-based Evidence Quality Reward (EQR) enforces that notes logically support the final answer, reducing distractor-induced errors.

Table 1: Performance comparisons on out-of-domain (TriviaQA, 2Wiki, Bamboogle, Musique, PopQA) and in-domain (NQ, HotpotQA) benchmarks. For each dataset, the **bold** indicates the best performance, and underline indicates the second-best performance.

| Methods | TriviaQA | | 2Wiki | | Bamboogle | | Musique | | PopQA | | NQ | | HotpotQA | |
|---|---|---|---|---|---|---|---|---|---|---|---|---|---|---|
| | F1 | EM | F1 | EM | F1 | EM | F1 | EM | F1 | EM | F1 | EM | F1 | EM |
| **Foundational Model** | | | | | | | | | | | | | | |
| Direct Inference | 0.321 | 0.298 | 0.264 | 0.228 | 0.221 | 0.216 | 0.085 | 0.074 | 0.170 | 0.150 | 0.198 | 0.134 | 0.244 | 0.216 |
| RAG Workflow | 0.456 | 0.442 | 0.244 | 0.232 | 0.254 | 0.240 | 0.100 | 0.094 | 0.479 | 0.458 | 0.420 | 0.376 | 0.371 | 0.330 |
| **CoT** | | | | | | | | | | | | | | |
| ReCAT | 0.474 | 0.438 | 0.482 | 0.336 | 0.272 | 0.184 | 0.192 | 0.118 | 0.367 | 0.308 | 0.495 | 0.454 | 0.421 | 0.380 |
| IRCoT | 0.432 | 0.418 | 0.492 | 0.417 | 0.245 | 0.112 | 0.192 | 0.102 | 0.322 | 0.287 | 0.512 | 0.470 | 0.435 | 0.392 |
| **Prompt-Based Agentic RAG** | | | | | | | | | | | | | | |
| Self-Ask | 0.392 | 0.362 | 0.336 | 0.278 | 0.332 | 0.320 | 0.260 | 0.214 | 0.410 | 0.398 | 0.471 | 0.423 | 0.410 | 0.365 |
| Iter-RetGen | 0.374 | 0.356 | 0.326 | 0.270 | 0.232 | 0.160 | 0.178 | 0.118 | 0.376 | 0.348 | 0.442 | 0.395 | 0.390 | 0.342 |
| Self-RAG | 0.451 | 0.436 | 0.432 | 0.391 | 0.351 | 0.256 | 0.192 | 0.183 | 0.332 | 0.314 | 0.508 | 0.465 | 0.448 | 0.402 |
| CR-Planner | 0.417 | 0.403 | 0.473 | 0.452 | 0.434 | 0.304 | 0.271 | 0.202 | 0.351 | 0.350 | 0.520 | 0.482 | 0.452 | 0.405 |
| Search-o1 | 0.589 | 0.566 | 0.286 | 0.272 | 0.358 | 0.328 | 0.168 | 0.140 | 0.369 | 0.336 | 0.345 | 0.310 | 0.330 | 0.268 |
| **RL-Based Agentic RAG** | | | | | | | | | | | | | | |
| ReARTeR | 0.468 | 0.506 | **0.554** | **0.534** | 0.119 | 0.096 | 0.296 | 0.237 | 0.432 | 0.422 | 0.545 | 0.502 | 0.512 | 0.465 |
| R1-Searcher | 0.731 | 0.688 | 0.491 | 0.446 | 0.201 | 0.176 | 0.228 | 0.214 | 0.427 | 0.413 | 0.538 | 0.492 | 0.498 | 0.451 |
| Search-R1 | 0.754 | 0.694 | 0.280 | 0.244 | 0.377 | 0.320 | 0.274 | 0.184 | **0.498** | **0.482** | 0.550 | 0.508 | 0.464 | 0.420 |
| **EviNote-RAG (Ours)** | **0.795** | **0.730** | 0.536 | 0.494 | **0.528** | **0.424** | **0.336** | **0.240** | 0.491 | 0.480 | **0.563** | **0.524** | **0.557** | **0.490** |

Overall, EVINOTE-RAG delivers consistent improvements across both in-domain and out-of-domain settings. These results demonstrate that effective noise filtering, coupled with reward design, enhances not only stability and accuracy but also generalization across diverse QA tasks. The Ablations in the following section further confirm that **SEN+EQR** is the strongest configuration across OOD sets while controlling sequence length and token usage, aligning with our generalization claim.

## 4.6 ABLATION STUDY

**Settings.** Our experiments build on **Base** model SEARCH-R1 (Jin et al., 2025), an end-to-end RAG pipeline trained with reinforcement learning. On top of this baseline, we examine several configurations. In the **Force Summary (FS)** setting, the model must output an explicit `<summary>` after each retrieval, with zero reward assigned if the tag is absent, which ensures strict compliance but increases reward sparsity. Relaxing this constraint, the **Naive Summary (NS)** setting allows the model to generate a concise `<summary>` of retrieved documents before answering, and the model still receives reward for a correct answer even when a summary is not provided. Building on NS, the **Supportive-Evidence Notes (SEN)** configuration enriches the summaries with evidence-aware annotations, improving focus and reducing misleading reasoning. Finally, the **SEN+EQR** configuration extends SEN by introducing the Evidence Quality Reward (EQR), which uses an Entail Judge to assess the quality of SENs and further enhance reasoning accuracy.

Overall, as shown in Tab. 2, the effectiveness of our workflow and training strategy makes SEN and SEN+EQR highly competitive, with performance consistently ranked as follows: SEN+EQR > SEN > Naive Summary > Base > Force Summary. Additionally, we observed the following experimental observations:

**Effectiveness of Dynamic Summarization.** Force Summary yields inferior results, indicating that rigid structural constraints and reward sparsity hinder model adaptability. In contrast, Naive Summary significantly outperforms the baseline, demonstrating that flexible summarization improves reasoning quality. In addition, we find that changing prompts to require summarization or evidence selection does not affect model performance (see Appendix A).

**Effectiveness of Evidence-aware Annotations:** The improvement observed when transitioning from Naive Summary to SEN validates our central hypothesis: the structured organization of evidence, coupled with explicit uncertainty markings (key info, uncertain info), serves as an efficient

Table 2: Ablation results on in-domain and out-of-domain QA benchmarks. Bold highlights the best performance, while underline marks the second best.

| Methods | TriviaQA | | 2Wiki | | Bamboogle | | Musique | | PopQA | | NQ | | HotpotQA | |
|---|---|---|---|---|---|---|---|---|---|---|---|---|---|---|
| | F1 | EM | F1 | EM | F1 | EM | F1 | EM | F1 | EM | F1 | EM | F1 | EM |
| Base | 0.754 | 0.694 | 0.280 | 0.244 | 0.377 | 0.320 | 0.274 | 0.184 | 0.498 | 0.482 | 0.550 | 0.508 | 0.464 | 0.420 |
| + Force Summary | 0.708 | 0.638 | 0.334 | 0.304 | 0.338 | 0.224 | 0.274 | 0.184 | 0.472 | 0.454 | 0.505 | 0.450 | 0.423 | 0.362 |
| + Naive Summary | 0.774 | 0.712 | 0.462 | 0.424 | 0.496 | 0.384 | 0.280 | 0.204 | 0.500 | 0.488 | 0.551 | 0.502 | **0.560** | **0.498** |
| + SEN | **0.795** | **0.730** | 0.505 | 0.464 | 0.440 | 0.352 | 0.317 | 0.210 | **0.524** | **0.514** | **0.563** | 0.518 | 0.550 | 0.482 |
| + SEN + EQR (Ours) | **0.795** | **0.730** | **0.536** | **0.494** | **0.528** | **0.424** | **0.336** | **0.240** | 0.491 | 0.480 | **0.563** | **0.524** | 0.557 | 0.490 |

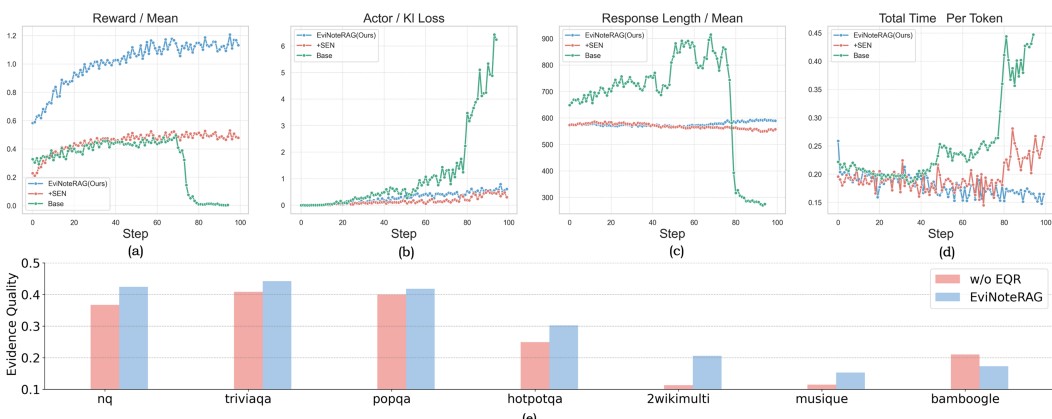

Figure 3: Training dynamics illustrating (a) reward, (b) KL Loss, (c) Response Length, and (d) Total Time Per Token (TPT). (e) Ablation study on EQR experiments.

filter for noise and enhances multi-hop reasoning accuracy. SEN's effectiveness arises from its dual capacity for selective evidence highlighting and uncertainty quantification.

**Effectiveness of EQR**    SEN+EQR achieves optimal performance through entailment-based supervision. The Evidence Quality Reward ensures logical consistency between generated notes and final answers, providing crucial semantic alignment that complements SEN's structural guidance. We further demonstrate the advantages of this setting in the supplementary material, showing that it guides the model toward thorough and effective summarization behavior while maintaining stability (Appendix A, B, and C).

## 4.7 TRAINING STABILITY AND PERFORMANCE ENHANCEMENT ANALYSIS

**Stable Training Requires Proper Workflow Design.**    Fig. 3(a)–(d) highlights the crucial role of workflow design in training stability. The base model (Jin et al., 2025) collapses around epoch 80, marked by rising KL divergence, declining rewards, and unstable actor loss. In contrast, EVINOTE-RAG adopts a retrieval–note–answer workflow that yields consistently stable curves across all metrics. By introducing structured instructions that resemble human note-taking, it reduces task difficulty, avoids degenerate outputs, and enables more stable policy learning.

**Behavioral Supervision and Noise Filtering Improve Efficiency.**    Fig. 3(c) shows that incorporating evidence-quality rewards (EQR) prevents unstable performance and excessive per-token latency. This improvement arises from EVINOTE-RAG's ability to produce shorter, more focused responses by filtering irrelevant content early, thereby reducing redundant reasoning. Fig. 3(d) further demonstrates that it maintains consistently lower token-level latency (Time Per Token, TPT), while the base model suffers from reasoning loops and collapses near epoch 80. Overall, the joint use of evidence note-taking and behavioral supervision yields shorter responses, reduces unnecessary computation, and enhances stability.

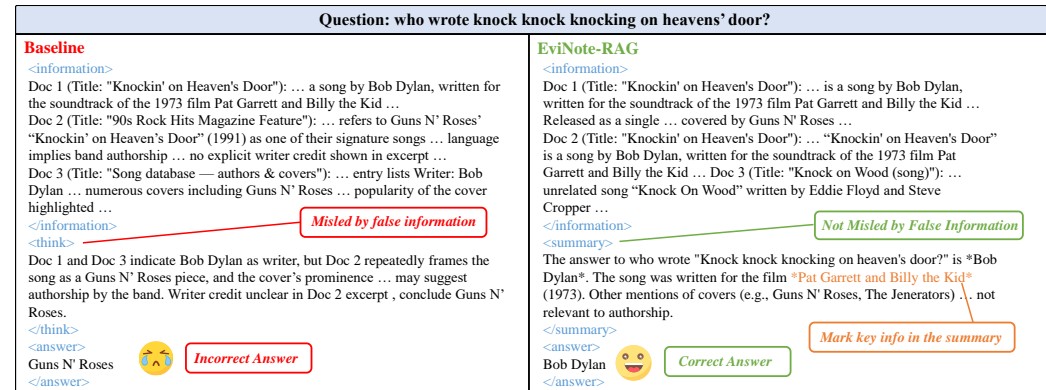

Figure 4: Case study on the query "who wrote *Knocking' on Heaven's Door*?". The baseline model is misled by misleading contextual information (Doc 2 repeatedly frames the song as a Guns N' Roses piece), resulting in the incorrect answer "Guns N' Roses". In contrast, our **EviNote-RAG** model effectively filters out misleading signals, emphasizes key evidence (e.g., writer credit in Doc 1 and Doc 2), and produces the correct answer "Bob Dylan". This highlights the importance of mitigating the interference of false or misleading information in knowledge-intensive tasks.

**EQR Improves SEN Quality Throughout Training.** Fig. 3(a) shows that the Evidence Quality Reward (EQR) steadily increases as training proceeds, indicating that the model learns to generate higher-quality Supporting Evidence Notes (SEN). This dynamic growth reflects the model's improved ability to align evidence with the target answer. In addition, Fig. 3(e) demonstrates that incorporating EQR leads to SEN with stronger entailment support compared to the ablated variant, highlighting the effectiveness of behavioral supervision. Together, these results confirm that EQR not only stabilizes training but also directly enhances the reasoning quality of SEN.

## 4.8 CASE STUDY

**Baseline Fails by Mixing Speculation with Evidence.** In the case in Fig. 4, the baseline retrieves passages noting that Knockin' on Heaven's Door was performed by Guns N' Roses but fails to distinguish between performance and authorship. Its reasoning chain introduces speculative guesses (e.g., "may suggest"), which dilute the role of explicit evidence in Doc 1 and Doc 3 stating that Bob Dylan wrote the song. As a result, the model wastes tokens on noisy deliberations and incorrectly concludes that Guns N' Roses are the authors.

**EviNote-RAG Clarify Evidence and Improve Efficiency.** EVINOTE-RAG highlights decisive authorship evidence (e.g., "Bob Dylan") in key-info notes (*), keeping reasoning constrained to facts directly relevant to "who wrote . . . ". Moreover, the Evidence Quality Reward (EQR) guides the model to produce clearer, entailment-supported notes that isolate answer-supportive information before generation; this yields shorter answers and lower token-level latency with stable decoding. Overall gains stem from evidence shaping (SEN+EQR). More case studies are provided in Appendix D.

## 5 CONCLUSION

We present **EviNote-RAG**, a framework that introduces a note-taking step to extract answer-supportive evidence before answering. By training LLMs to produce *Supportive-Evidence Notes (SENs)* and guiding them via a tailored entailment-based reward, our approach improves answer accuracy. Extensive experiments demonstrate that EviNote-RAG achieves state-of-the-art performance while enhancing training stability. These results highlight the benefits of evidence-focused abstraction for robust, faithful retrieval-augmented reasoning. Beyond empirical gains, our work establishes a general recipe for integrating structured note-taking with reward design, offering a principled path toward more interpretable and controllable RAG systems.

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

LLM USAGE STATEMENT

In the preparation of this manuscript, a Large Language Model (LLM) was employed solely for language polishing. All academic content, interpretations, and responsibilities remain entirely with the authors.

## A  EFFECT OF SUMMARY STRATEGIES

Table 3: Ablation study on in-domain and out-of-domain QA tasks. We compare different instructional designs: Naive Summary (NS), Naive Evidence (NE), Force Summary (FS), and our proposed Supportive-Evidence Notes (SEN). **Bold** indicates the best performance, while underline marks the second best.

| Methods | TriviaQA | | 2Wiki | | Bamboogle | | Musique | | PopQA | | NQ | | HotpotQA | |
|---|---|---|---|---|---|---|---|---|---|---|---|---|---|---|
| | F1 | EM | F1 | EM | F1 | EM | F1 | EM | F1 | EM | F1 | EM | F1 | EM |
| Base | 0.754 | 0.694 | 0.280 | 0.244 | 0.377 | 0.320 | 0.274 | 0.184 | 0.498 | 0.482 | 0.550 | 0.508 | 0.464 | 0.420 |
| + NS | 0.774 | 0.712 | 0.462 | 0.424 | **0.496** | **0.384** | 0.280 | 0.204 | 0.500 | 0.488 | 0.551 | 0.502 | **0.560** | **0.498** |
| + NE | 0.773 | 0.710 | 0.460 | 0.422 | 0.494 | 0.382 | 0.281 | 0.206 | 0.501 | 0.486 | 0.552 | 0.503 | 0.559 | 0.497 |
| + FS | 0.708 | 0.638 | 0.334 | 0.304 | 0.338 | 0.224 | 0.274 | 0.184 | 0.472 | 0.454 | 0.505 | 0.450 | 0.423 | 0.362 |
| + SEN | **0.795** | **0.730** | **0.505** | **0.464** | 0.440 | 0.352 | **0.317** | **0.210** | **0.524** | **0.514** | **0.563** | **0.518** | 0.550 | 0.482 |

### A.1  EXPERIMENTAL SETUP FOR SUMMARY STRATEGY ABLATIONS

We evaluate summary strategies using the unified QA setup described in the *Experiments* section (Section 4). Unless otherwise noted, all components (retriever, datasets, metrics) and hyperparameters follow the main setup to control for confounding factors. The base system (Jin et al., 2025) implements an end-to-end reinforcement learning pipeline for retrieval-augmented generation (RAG), upon which we vary only the summary strategy as follows:

- **Naive Summary (NS)**: The model can dynamically generate a concise `<summary>` of the retrieved documents prior to answering, without incorporating evidence-aware annotations. Unlike FS setting, the model remains eligible for a reward if the answer is correct, even when a summary is not provided.

- **Naive Evidence (NE)**: A tag-variant of NS in which the model is instructed to output an `<evidence>` section instead of `<summary>` and to include only answer-relevant content. Beyond the tag replacement and this content restriction, the procedure mirrors NS and introduces no additional supervision.

- **Force Summary (FS)**: The model is forced to produce an explicit `<summary>` section after each retrieval. If the tag is absent, the reward is set to zero during training, which strictly enforces compliance but consequently increases reward sparsity.

- **Supportive-Evidence Notes (SEN)**: Our proposed strategy that guides the model to extract and organize supporting evidence into structured notes before answering. SEN further requires explicit marking of *key* information (with "$\star$") and *uncertain* information (with "$-$"), promoting fine-grained supervision aligned with human note-taking.

### A.2  MAIN RESULTS

As shown in Tab. 3, our ablation results provide a systematic comparison across different instructional designs. Several consistent patterns emerge.

**Overall Ranking of Instruction Designs.**    A clear hierarchy of effectiveness can be observed:
$$\text{SEN} > \text{NS} \approx \text{NE} > \text{Base} > \text{FS}.$$
This ranking reflects the strength of supervision each design introduces. SEN enforces structured note-taking and yields the most effective, high–information-utilization summary; NS and NE provide only weak summarization signals; Base relies purely on raw retrieval without additional supervision, while FS over-constrains optimization with sparse rewards and ultimately harms performance. These results show that:

> **Finding 1.** Effective summary strategies are not about requiring or forcing summaries, but about organizing supportive evidence to meaningfully guide reasoning.

**SEN vs. NS vs. FS.** Compared with naive summaries (NS), SEN encourages explicit identification and organization of supporting evidence, rather than compressing retrievals into a single passage. This design substantially reduces noise from irrelevant documents and better aligns the model's reasoning with human note-taking practices. By contrast, FS enforces summary production too rigidly, introducing instability during optimization and degrading performance. Together, these results highlight the importance of instructional flexibility combined with evidence structuring, which SEN uniquely achieves.

**Effect of Evidence Marking.** We further disentangle whether improvements stem from mere tag changes or from structural constraints. Simply replacing the `<summary>` tag with `<evidence>` and requiring evidence-only summaries yields virtually identical performance to NS, i.e., `evidence ≈ summary`. However, when we further require explicit evidence marking (i.e., SEN), the model benefits significantly. This suggests that:

> **Finding 2.** Gains come from highlighting key evidence. Methods that structure evidence (selection, organization, explicit marking) outperform those that only require a summary format, yielding stronger factual accuracy.

### A.3 TRAINING DYNAMIC ANALYSIS.

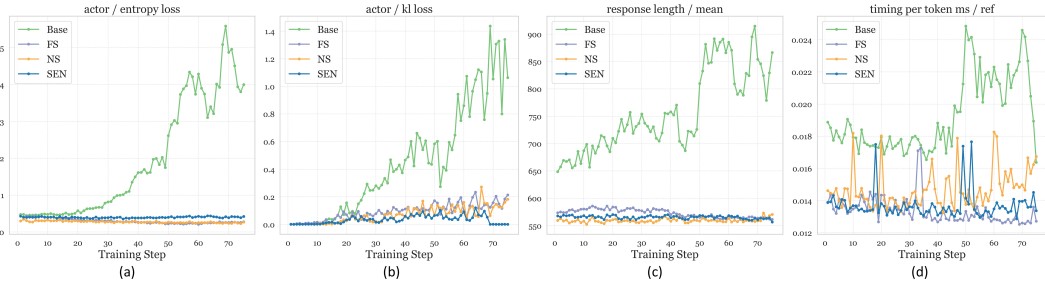

Figure 5: **Training dynamics under different summary strategies.** (a) actor entropy loss; (b) actor KL loss (w.r.t. the reference policy); (c) mean response length; (d) token-level latency (ms/token). SEN maintains low entropy and KL drift with stable, shorter responses and low latency; NS is slightly less stable but similar in trend; FS achieves low latency at the cost of under-exploration and weaker accuracy; the BASE policy exhibits late-stage blow-up in KL/entropy, response-length sprawl, and higher per-token latency.

**Stability (Fig. 5a–b).** SEN yields the most stable optimization: policy entropy and KL divergence remain low and flat across training, indicating controlled exploration and limited drift from the reference policy. NS shows a similar but slightly noisier profile. By contrast, the BASE policy exhibits a late-stage surge in both entropy and KL, signalling distribution shift and unstable updates. FS keeps KL small but does not translate this regularization into accuracy improvements, consistent with under-exploration caused by rigid compliance constraints.

**Efficiency (Fig. 5c–d).** SEN/NS produce consistently shorter, more focused responses (hundreds of tokens fewer than BASE) and sustain low ms/token latency. The BASE policy's response length inflates markedly in later steps, accompanied by a clear rise in per-token latency. FS attains the lowest latency overall, but its gains reflect conservative decoding rather than improved reasoning, aligning with its inferior task performance.

**Overall Analysis.** The curves corroborate our ablation ranking (SEN > NS ≈ NE > Base > FS): The SEN stabilizes optimization (low KL/entropy), filters noise to keep responses concise, and improves runtime efficiency—benefits that rigidly enforced summaries (FS) fail to realize. These observations are consistent with the training-stability analysis reported in the paper, where structured supervision densifies useful reward signals and regularizes the policy towards faithful evidence use.

## B  ANALYSIS OF REWARD SHAPING APPROACHES

Table 4: **Reward shaping ablations**. We compare Force Summary (FS) and its shaped variants with Stochastic Reward (SR) and Evidence Quality Reward (EQR), and contrast them with Supportive-Evidence Notes (SEN) and SEN+EQR. SR serves as a stochastic control to test whether gains come from reward perturbations alone; **SEN** and **EQR** are our proposed components. **Bold** marks the best performance; underline marks the second best.

| Methods | TriviaQA | | 2Wiki | | Bamboogle | | Musique | | PopQA | | NQ | | HotpotQA | |
|---|---|---|---|---|---|---|---|---|---|---|---|---|---|---|
| | F1 | EM | F1 | EM | F1 | EM | F1 | EM | F1 | EM | F1 | EM | F1 | EM |
| Base | 0.754 | 0.694 | 0.280 | 0.244 | 0.377 | 0.320 | 0.274 | 0.184 | 0.498 | 0.482 | 0.550 | 0.508 | 0.464 | 0.420 |
| + FS | 0.708 | 0.638 | 0.334 | 0.304 | 0.338 | 0.224 | 0.274 | 0.184 | 0.472 | 0.454 | 0.505 | 0.450 | 0.423 | 0.362 |
| + FS + SR | 0.752 | 0.690 | 0.410 | 0.398 | 0.328 | 0.296 | 0.276 | 0.188 | 0.514 | 0.502 | 0.546 | 0.500 | 0.460 | 0.402 |
| + FS + EQR | 0.766 | 0.704 | 0.426 | 0.414 | 0.224 | 0.208 | 0.274 | 0.184 | **0.528** | **0.518** | 0.551 | 0.510 | 0.464 | 0.408 |
| + SEN | **0.795** | **0.730** | 0.505 | 0.464 | 0.440 | 0.352 | 0.317 | 0.210 | 0.524 | 0.514 | **0.563** | 0.518 | 0.550 | 0.482 |
| + SEN + EQR | **0.795** | **0.730** | **0.536** | **0.494** | **0.528** | **0.424** | **0.336** | **0.240** | 0.491 | 0.480 | **0.563** | **0.524** | **0.557** | **0.490** |

### B.1  EXPERIMENTAL SETUP FOR REWARD SHAPING

In our experimental design, we retain the unified QA framework introduced in the previous section and vary only the reward signals. This allows us to isolate the effect of different reward shaping strategies while keeping the model architecture and training pipeline fixed. In addition to the baseline (Base), Force Summary (FS), and Supportive-Evidence Notes (SEN), we introduce two further reward mechanisms:

- **Stochastic Reward (SR)**: A mechanism that provides a small reward (0.1) with probability $1/3$ even when the predicted answer is incorrect. The motivation is to alleviate reward sparsity under FS during early training, preventing the model from stagnating due to overly strict zero-reward penalties.

- **Evidence Quality Reward (EQR)**: Our entailment-based reward function, which evaluates whether the final note (or summary) semantically supports the gold answer. By explicitly encouraging consistency between retrieved evidence and the correct answer, EQR not only mitigates reward sparsity but also directly aligns the optimization process with the task objective of evidence-faithful reasoning.

*Note:* Among all variants, **SEN** and **EQR** are our proposed components.

### B.2  MAIN RESULTS: FS VS. SR VS. EQR

**Overall ranking.** Across all datasets, as shown in Tab. 4, the methods follow a clear hierarchy:

$$\text{SEN+EQR} > \text{SEN} > \text{FS+EQR} > \text{FS+SR} = \text{BASE} > \text{FS}.$$

This ordering highlights two key insights: (1) semantic alignment through EQR improves over purely stochastic shaping (SR), but (2) structural supervision (SEN) is essential, as it consistently delivers the largest performance gains.

**SEN remains the primary driver of performance.** Structural supervision from SEN delivers the strongest improvements across almost all benchmarks. By guiding the model to explicitly organize supportive evidence, SEN alleviates reward sparsity. Even without additional shaping, SEN alone surpasses all FS-based methods.

**SEN+EQR achieves the best overall results.** The combination of structural supervision (SEN) and semantic shaping (EQR) provides the best balance of stability and task alignment. SEN+EQR consistently outperforms both standalone SEN and FS-based variants, achieving the strongest results on 2Wiki, Bamboogle, and Musique, while maintaining top-tier performance on TriviaQA, NQ, and HotpotQA.

**FS alone degrades performance.** Using FS in isolation leads to degraded performance. Since FS enforces the `<summary>` structure through zero-reward penalties, it introduces severe reward sparsity. This "hard penalty" discourages exploration and limits the model's ability to discover useful behaviors, resulting in accuracy that often falls below the BASE model on several datasets.

**SR partially alleviates sparsity but lacks semantic guidance.** Introducing SR helps smooth the optimization process by reducing the harshness of reward sparsity. By occasionally rewarding incorrect answers, SR enables more stable training and closes part of the gap to BASE. However, the gains remain modest because SR is not semantically aligned: the reward does not provide guidance about evidence faithfulness, leaving the model largely uninformed about whether its reasoning supports the gold answer.

**EQR provides task-aligned shaping; SEN remains essential.** Replacing SR with the entailment-based EQR yields larger, more consistent improvements over FS. However, structural supervision from SEN remains the primary driver: SEN surpasses all FS-based variants, and combining EQR with SEN achieves the best overall results across benchmarks (SEN+EQR).

> **Finding 3.** Reward shaping is most effective when semantically aligned with evidence quality and paired with structured supervision: EQR improves over FS/FS+SR, but SEN+EQR delivers the strongest and most consistent gains across datasets.

## C  JOINT EFFECTS AND ADVANCED ANALYSIS

Table 5: Ablation study on in-domain and out-of-domain QA tasks. We report the effect of Force Summary (FS), Stochastic Reward (SR), Supportive-Evidence Notes (SEN), and Evidence Quality Reward (EQR). SR serves as a stochastic control to examine whether improvements stem from reward perturbations alone, while SEN and EQR represent our proposed modules. **Bold** highlights the best performance, while underline marks the second best.

| Methods | TriviaQA | | 2Wiki | | Bamboogle | | Musique | | PopQA | | NQ | | HotpotQA | |
|---------|------|------|------|------|------|------|------|------|------|------|------|------|------|------|
| | F1 | EM | F1 | EM | F1 | EM | F1 | EM | F1 | EM | F1 | EM | F1 | EM |
| Base | 0.754 | 0.694 | 0.280 | 0.244 | 0.377 | 0.320 | 0.274 | 0.184 | 0.498 | 0.482 | 0.550 | 0.508 | 0.464 | 0.420 |
| + NS | 0.774 | 0.712 | 0.462 | 0.424 | 0.496 | 0.384 | 0.280 | 0.204 | 0.500 | 0.488 | 0.551 | 0.502 | **0.560** | **0.498** |
| + FS | 0.708 | 0.638 | 0.334 | 0.304 | 0.338 | 0.224 | 0.274 | 0.184 | 0.472 | 0.454 | 0.505 | 0.450 | 0.423 | 0.362 |
| + FS + SR | 0.752 | 0.690 | 0.410 | 0.398 | 0.328 | 0.296 | 0.276 | 0.188 | 0.514 | 0.502 | 0.546 | 0.500 | 0.460 | 0.402 |
| + FS + EQR | 0.766 | 0.704 | 0.426 | 0.414 | 0.224 | 0.208 | 0.274 | 0.184 | **0.528** | **0.518** | 0.551 | 0.510 | 0.464 | 0.408 |
| + SEN | **0.795** | **0.730** | 0.505 | 0.464 | 0.440 | 0.352 | 0.317 | 0.210 | 0.524 | 0.514 | **0.563** | 0.518 | 0.550 | 0.482 |
| + SEN + EQR | **0.795** | **0.730** | **0.536** | **0.494** | **0.528** | **0.424** | **0.336** | **0.240** | 0.491 | 0.480 | **0.563** | **0.524** | 0.557 | 0.490 |

### C.1  EXPERIMENTAL SETTINGS FOR ABLATION STUDY

We follow the same unified QA setup and training protocol described in the previous section. Unless otherwise noted, model architecture, optimization schedule, and data splits remain unchanged. The only differences across variants lie in the instruction strategies and reward signals, ensuring that observed effects can be attributed solely to the proposed modules (SEN and EQR) or their ablations.

These templates define the behavior of the agent under different summary strategies, and their design choices directly account for the performance differences observed in our ablation studies.

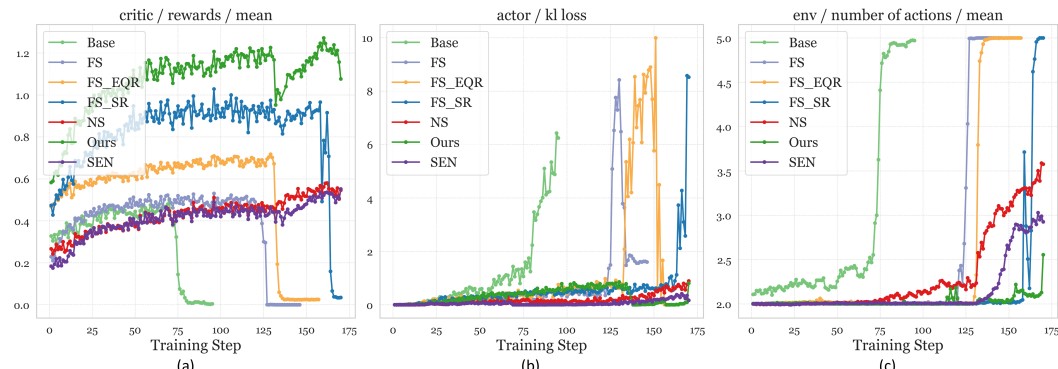

Figure 6: **Training stability.** Actor-side stability diagnostics across methods (Base, FS, FS+SR, FS+EQR, NS, SEN, and **Ours** = SEN+EQR). Panels: (a) reward score, (b) actor KL loss w.r.t. the reference policy. Lower and smoother curves indicate more stable optimization. (c) number of actions. When the model generates invalid actions, it tends to repeat the previous behavior, leading to a rapid increase in action frequency.

C.2    BEST PERFORMING COMBINATIONS (E.G., SEN+EQR).

**Synergy Analysis of Instruction and Reward.**    EQR encourages the model to produce higher-quality evidence, improving factual reliability. On its own, EQR provides moderate gains, but the strongest improvements arise when it is combined with SEN. This synergy enables the model not only to identify relevant evidence but also to prioritize higher-quality reasoning chains, leading to the best performance across both in-domain and out-of-domain QA tasks.

**In-domain vs. Out-of-domain Generalization.**    A closer look at Tab. 5 reveals that SEN alone already establishes strong in-domain gains on factoid QA datasets such as HotpotQA and NQ. However, the addition of EQR becomes particularly impactful in out-of-domain or compositional settings, such as 2Wiki, Bamboogle, and Musique, where SEN+EQR consistently achieves the highest F1 and EM. This indicates that semantic shaping through EQR plays a critical role in transferring structural supervision to unfamiliar domains.

**Comparison against Naive Summarization (NS).**    The contrast between SEN and instructional naive summarization (NS) underscores the importance of structured evidence organization. While NS sometimes improves over the base model, its benefits are inconsistent, and it occasionally introduces noise by forcing the model to compress evidence prematurely. By contrast, SEN's explicit structuring yields consistent improvements across all datasets, confirming that inductive biases toward evidence organization are more effective than general summarization prompts.

**Overall Patterns.**    Bringing these observations together, we can summarize the joint effects as follows:

$$\text{SEN+EQR} > \text{SEN} > \text{NS} > \text{FS+EQR} > \text{BASE} > \text{FS+SR} > \text{FS}.$$

This ordering highlights two key findings: (1) SEN is indispensable as the structural backbone of our framework, and (2) the benefits of EQR are most pronounced when paired with SEN, enabling robust generalization to both factoid and multi-hop QA tasks.

C.3    TRAINING STABILITY AND EFFICIENCY

**Stability (Fig. 6).**    Overall, we observe different degrees of collapse across the control variants. In terms of collapse order, the stability ranking is:

$$\text{OURS} \approx \text{NS} \approx \text{SEN} > \text{FS+SR} > \text{FS+EQR} > \text{FS} > \text{BASE}.$$

This ordering highlights that structural supervision (SEN) is the key factor preventing collapse, while stochastic shaping (SR) or entailment alignment (EQR) alone provide only partial stabilization. In addition, **SEN** and **Ours** maintain low, smooth entropy and KL throughout training, together

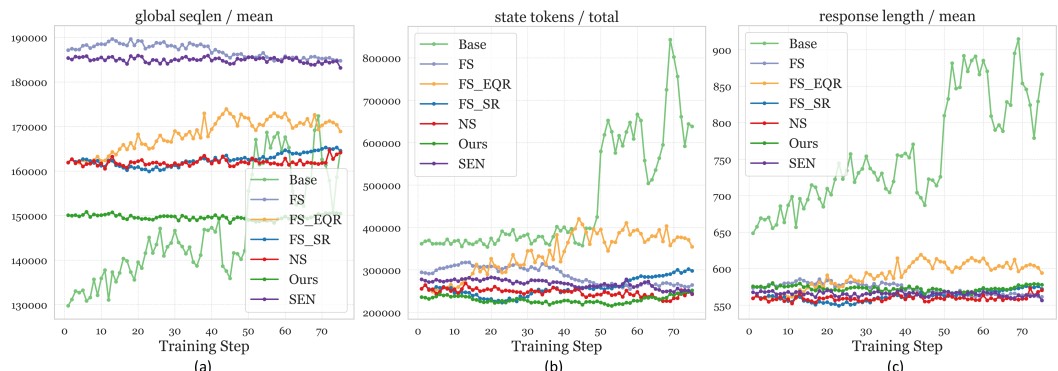

Figure 7: Training efficiency analysis across methods (Base, FS, FS+SR, FS+EQR, NS, SEN, and Ours=SEN+EQR). (a) Average global sequence length, (b) total state tokens, and (c) mean response length. While the Base policy suffers from uncontrolled length growth and instability, FS variants suppress length but often reflect conservative decoding without accuracy gains. By contrast, SEN and SEN+EQR maintain concise, stable responses with lower variance, confirming that structured supervision and semantic shaping jointly improve efficiency.

with steadily improving rewards, indicating controlled exploration and limited drift from the reference policy. In contrast, **Base** exhibits a late-stage surge in KL, accompanied by reward collapse, revealing a distribution shift under noisy retrieval. **FS** keeps KL small but fails to convert this regularization into accuracy due to zero-reward penalties that induce under-exploration. Adding **SR** partially alleviates sparsity by smoothing the reward landscape, but gains remain limited because it lacks semantic alignment with evidence quality. In comparison, the entailment-aligned **EQR** produces both more stable reward trajectories and higher peaks; when combined with **SEN** (**Ours**), it delivers the most robust and consistent improvements across datasets. Furthermore, Fig. 6(c) reports the *number of actions* executed during training. We find that when the model generates invalid actions, it often repeats the previous behavior, leading to a rapid escalation in action counts. This instability is especially pronounced under **FS**, where zero-reward penalties trigger repetitive failure modes. In contrast, **SEN+EQR** effectively regulates the action space, avoiding runaway repetitions and thereby ensuring more efficient and reliable policy updates.

**Training Efficiency (Fig. 7).** The efficiency curves complement our stability analysis by illustrating how different methods manage output length and token usage during training. First, the **Base** model exhibits pronounced growth in both global sequence length (Fig. 7a) and response length (Fig. 7c), often exceeding 900 tokens, which signals uncontrolled decoding and redundant reasoning. This sprawl aligns with the instability patterns observed in KL divergence and reward collapse, showing that the lack of structured supervision leads to degenerate behaviors.

In contrast, **FS** and its shaped variants constrain response length (often below 600 tokens), but this reduction largely reflects conservative decoding rather than improved reasoning, consistent with their weaker QA performance. **SR** provides partial smoothing of the optimization landscape, while **EQR** encourages more semantically grounded evidence use, leading to slightly shorter but more reliable responses.

**SEN** and especially **SEN+EQR (Ours)** achieve the best balance: they keep global sequence length and state tokens stable across training, while producing consistently concise and focused responses. This efficiency stems from structured evidence organization (SEN), which filters noise before generation, and semantic reward shaping (EQR), which discourages degenerate repetitions. Together, these mechanisms prevent runaway growth in sequence length and maintain efficient decoding, corroborating the overall finding that SEN+EQR delivers both stability and efficiency.

**Takeaways.** These dynamics corroborate our ablations: SEN > NS ≈ NE > Base > FS. Structuring and validating evidence (SEN + EQR) densifies useful reward signals and regularizes the policy toward evidence-faithful reasoning, yielding both *greater stability* and *better efficiency*.

---

**Finding 4.** Semantic reward shaping alone (EQR) yields more stable training and improves factual reliability compared to stochastic shaping, but its benefits are limited without structural guidance. When combined with SEN, the two components act synergistically, producing the most consistent gains in both accuracy and generalization across QA benchmarks.

---

**Question: who won the oscar for best picture in 1976?**

**Baseline**

<information>     *Too Much Noisy Information*

Doc 1 (Title: "48th Academy Awards")
Mentions the ceremony held March 29, 1976, but does not clearly state the Best Picture winner. Describes films like Jaws and "Traffic", and lists general details about nominations and awards.
Doc 2 (Title: "48th Academy Awards")
Confirms the 48th Academy Awards took place in Los Angeles, 1976, hosted by Walter Matthau, Robert Shaw, George Segal, Goldie Hawn, and Gene Kelly. Provides context about broadcasting and side events, but again no explicit Best Picture winner.
Doc 3 (Title: "12th Academy Awards")
Historical context from 1940 (12th Awards). Mentions films like The Wizard of Oz, Ninotchka, Of Mice and Men, and introduces the first Academy Award for special effects. Not relevant for 1976 Best Picture.
</information>
<think>     *Reasoning Inconsistent with the Question*

"Network" was the winner of the Oscar for Best Picture in 1976. Other details about the awards ceremony, including presenters and categories, were also mentioned but were not relevant to the question.
</think>
<answer>     *Incorrect Answer*
Network
</answer>

**EviNote-RAG**

<information>
Doc 1 (Title: "49th Academy Awards") The 49th Academy Awards ceremony (held in 1977, honoring 1976 films) awarded Best Picture to "Rocky". Other major winners included "Network" in acting/writing categories.
Doc 2 (Title: "48th Academy Awards") The 48th Academy Awards were held on March 29, 1976, honoring 1975 films. Not directly about 1976 films.
Doc 3 (Title: "Network (1976 film)") "Network" won multiple Oscars (acting/writing) for the 1976 film year, but not Best Picture.
</information>
<summary>     *Filter Noise , Summarize , Identify Intent*

The answer to who won the Oscar for Best Picture in 1976 is *"Rocky."* Additionally, the information mentions "Beginners" and "Network", but the query specifically asks who won Best Picture in 1976.
</summary>
<answer>     *Correct Answer*     *Further mark key info in the summary*
Rocky
</answer>

**Highlight: More accurate intent recognition**

Figure 8: **Temporal-intent disambiguation in open-domain QA.** Query: "Who won the Oscar for Best Picture in 1976?" The baseline retrieves and reasons over noisy passages about the *48th Academy Awards* (1976 ceremony; honoring 1975 films) and predicts *Network*. EVINOTE-RAG detects the temporal intent ("1976 films ⇒ *49th* Academy Awards, held in 1977"), filters distractors, highlights the key evidence, and correctly answers *Rocky*. Red callouts mark baseline failure modes (noise and off-intent reasoning); green callouts mark EVINOTE-RAG behaviors (filter, summarize, identify intent).

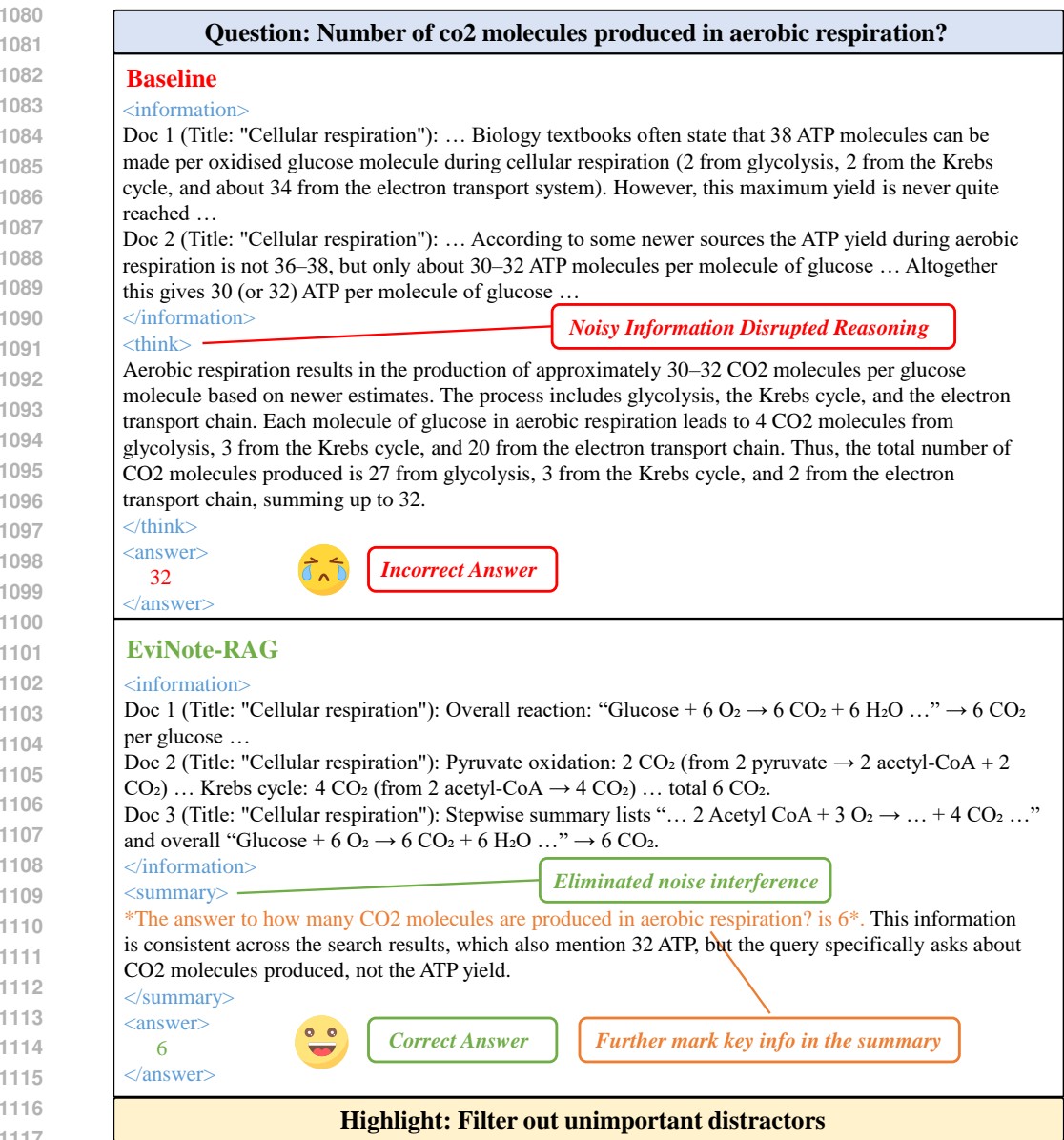

Figure 9: **Distractor suppression in scientific QA.** Query: "Number of $CO_2$ molecules produced in aerobic respiration?" The baseline conflates ATP yield facts and answers *32*. EVINOTE-RAG retains only reaction-relevant facts—*2 $CO_2$* from pyruvate oxidation and *4 $CO_2$* from the TCA cycle—totaling *6 $CO_2$* per glucose, and answers *6*. Red callouts indicate distractor-driven errors; green callouts show how EVINOTE-RAG filters noise and foregrounds the correct variable.

# D   MORE CASE STUDIES: INTENT RECOGNITION AND DISTRACTOR FILTERING WITH EVINOTE-RAG

We provide two qualitative case studies complementing the quantitative results. They show how EVINOTE-RAG improves answer accuracy by (i) recognizing user intent (especially temporal intent) and (ii) filtering distractors during retrieval-augmented reasoning. In both cases, EVINOTE-RAG and the baseline operate over comparable retrieved passages; the difference is that EVINOTE-RAG enforces a note-taking discipline with explicit `<information>` → `<summary>` → `<answer>` stages that compress, align, and verify evidence against the question.

### D.1 CASE A: TEMPORAL-INTENT DISAMBIGUATION (BEST PICTURE, 1976)

**Task & challenge.** Queries naming a year and an award admit two competing interpretations: the *ceremony year* vs. the *content year* (films produced in that year). Popular documents discuss both, making temporal intent easy to misread.

**Baseline behavior.** The baseline surfaces documents about the *48th Academy Awards* (held in 1976, honoring 1975 films) and the film *Network*, then produces an answer consistent with that off-intent thread of reasoning. Failure modes: (1) evidence overload—long verbatim passages not aligned to the asked year; (2) intent drift.

**EVINOTE-RAG behavior.** The `<information>` notes isolate short, query-conditioned facts, e.g., "49th Academy Awards (held 1977, honoring 1976 films); Best Picture: *Rocky*." The `<summary>` explicitly states the intent mapping ("1976 $\Rightarrow$ winners announced at the 49th ceremony") and marks the decisive evidence. This structured condensation suppresses ceremony-year distractors and yields the correct answer *Rocky*.

**Takeaways.** EviNote-style notes act as a temporal alignment layer: before generation, the system resolves event/year semantics; after alignment, plausible but off-intent documents lose influence.

### D.2 CASE B: DISTRACTOR SUPPRESSION ($CO_2$ MOLECULES IN AEROBIC RESPIRATION)

**Task & challenge.** The question asks for the *count of $CO_2$ molecules per glucose*. Corpus passages often co-mention ATP yields (about 30–32), a frequent but irrelevant number that can anchor the model.

**Baseline behavior.** The baseline mixes ATP-yield statements into its reasoning trace and outputs *32*, a distractor-anchoring error.

**EVINOTE-RAG behavior.** The `<information>` notes keep only stoichiometrically relevant facts: *2 $CO_2$* from pyruvate oxidation and *4 $CO_2$* from the Krebs/TCA cycle, matching the overall reaction "glucose $+ 6\,O_2 \rightarrow 6\,CO_2 + 6\,H_2O$." The `<summary>` restates the tally and reasserts the target variable ($CO_2$ count, not ATP). The final answer is *6*.

**Takeaways.** By forcing an intermediate summary that names the variable of interest and aggregates counts, EVINOTE-RAG resists frequent-but-irrelevant facts and prevents numeric anchoring.

**Overall observation.** Across both cases, improvements arise not from additional retrieval, but from *evidence shaping*: (1) extracting minimal, question-conditioned notes; (2) explicitly resolving intent (time, entity, variable); and (3) committing to a concise summary before answering. This mirrors the measured gains on both out-of-domain and in-domain benchmarks.

