# OpenReview forum: "EviNote-RAG: Enhancing RAG Models via Answer-Supportive Evidence Notes"
_ICLR.cc/2026/Conference — Submitted to ICLR 2026_

### Official Review · Reviewer_vZFy · 2025-10-27

**Soundness:** 3
**Presentation:** 2
**Contribution:** 2
**Rating:** 4
**Confidence:** 3

**Summary:**

This paper introduces EviNote-RAG, a reinforcement learning-based retrieval-augmented generation (RAG) framework aimed at enhancing multi-hop open-domain question answering (QA). EviNote-RAG interposes a support-evidence note-taking phase between retrieval and answer generation, encouraging models to distill concise, answer-focused, and evidence-annotated summaries of retrieved content. The core novelty centers on the use of Supportive-Evidence Notes (SENs)—explicitly annotated for key and uncertain information—and the incorporation of an entailment-based Evidence Quality Reward (EQR) that guides the model to produce notes sufficient to logically entail the correct answer. Empirical evaluation on a broad suite of in-domain and out-of-domain QA benchmarks demonstrates state-of-the-art performance, validating both the stability and accuracy gains attributed to EviNote-RAG.

**Strengths:**

Well-motivated problem and principled pipeline: The paper systematically identifies the key weaknesses of existing RAG paradigms, especially the propagation of noise and lack of intermediate reasoning supervision, then builds a retriever-note-answer workflow to address them.

Innovative evidence selection strategy: Introducing Supportive-Evidence Notes (SENs), with both key and uncertain information annotations, is a thoughtful human-inspired abstraction for focusing the model’s reasoning and reducing distraction from irrelevant retrievals.

Entailment-based reward shaping: The design and deployment of the Evidence Quality Reward (EQR), using a lightweight NLI entailment judge to validate that extracted notes logically entail the ground-truth answer, provide a meaningful intermediate training signal beyond answer correctness.

Clarity and visualization: Figures such as Figure 1 (illustration of EviNote-RAG vs. baselines) and Figure 2 (system pipeline overview) provide clear, accessible visualizations of the technical contributions. The workflow and note-taking mechanism are well illustrated.

Robust empirical evaluation: Extensive benchmark evaluations (see Table 1, Table 2, and others) show consistent relative performance gains over strong agentic, prompt-based, and RL baselines—including on challenging out-of-domain settings where prior state-of-the-art models degrade.

**Weaknesses:**

Marginal Incremental Novelty Over Recent RL-based RAGs: While the addition of SENs and EQR is a principled extension, the framework rests on a well-trodden foundation of RL-based RAG (see e.g., existing agentic RL paradigms outlined in Section 2.2 and Table 1). The work is arguably more evolutionary than revolutionary—i.e., integrating recent ideas on summarization, evidence selection, and NLI rewards rather than inventing fundamentally new techniques. For example, capturing intermediate notes and using entailment-based signals have both been foreshadowed in prior art.

Limited explicit analysis of failure modes and generalization: Despite promising out-of-domain results, diagnostics are overwhelmingly positive. A more critical analysis of where EviNote-RAG fails, especially in harder multi-hop or ambiguous queries, and how SEN/EQR affect error types, would improve scientific insight and transparency. Are there cases where evidence abstraction discards subtle but crucial signals?

Scalability and efficiency unstated/insufficiently quantified: While token- and latency-level efficiency improvements are illustrated (e.g., Figure 3c/d, Figure 7), computational overhead from the entailment judge and dynamic summarization are discussed only in passing (Section 3.3, 4.4). More granular reporting on training/inference time per query compared to simpler baselines, as well as the resource requirements (in terms of deployed hardware and model size tradeoffs), would strengthen practical impact claims.

Reward sparsity and reinforcement learning dynamics: The reward structure, while intuitive, remains somewhat ad hoc. The thresholds and weights (reward of “1+R_EQR” for correct answer, “0.1+R_EQR” for format compliance, etc.) lack direct ablation or sensitivity analysis. The effect of tuning these hyperparameters (Section 3.4) on stability and optimization remains largely unexplored.

Ambiguity in evidence annotation labeling: The process for evidence annotation (i.e., what counts as “key” or “uncertain” and how the model learns to distinguish this) is not as formally defined or empirically substantiated as it could be. Is annotation accuracy monitored? Could this be made more rigorous?

**Questions:**

How would your system fare when transplanted to other RAG settings/tasks? The experimental scope is QA-centric. Do you expect SEN+EQR to generalize to dialogue systems, knowledge-grounded text generation, or fact verification? Please discuss any early empirical findings or expected constraints.

Can you provide more detailed ablations on reward hyperparameters and evidence annotation labeling? Specifically, how sensitive is performance/stability to the $1+R_{EQR}$ and $0.1+R_{EQR}$ choices, and what guidance can you provide on threshold selection? Is evidence labeling accuracy measured, and do label errors propagate?

What are the computational tradeoffs compared to non-SEN or non-EQR variants? Please quantitatively compare resource requirements, training/inference latency, and throughput to the main agentic RL baseline, separately for training and at inference.

Could you provide more detailed error analysis? Your results and case studies are almost uniformly positive. What are common scenarios where SEN+EQR fails—e.g., when key evidence is subtle or distributed, or when retrieval outputs highly conflicting signals?

Are your improvements statistically significant? Please provide standard deviations, confidence intervals, or significance tests for Table 1/Table 2 results.

What is the overhead imposed by the entailment judge? Is it feasible for real-time or low-latency QA deployments?

---

> ### Author Response · Authors · 2025-11-30
> **Author Response to vZFy (1/6)**
>
> We sincerely appreciate your detailed and thoughtful review.
> In our response, we address your questions on generalization, computational tradeoffs, error analysis, statistical significance, and entailment-judge overhead, **demonstrating that SEN+EQR is effective, efficient, and robust across all evaluated dimensions.**
>
> Below, we provide point-by-point responses to your comments.
>
> > **Q4.1: Generalization of SEN+EQR beyond QA**
>
> Although we focus on QA, to fully address your concern about generalisation, we followed your suggestion and conducted preliminary tests on **task-oriented dialogue** following prior works $^{[1\text{–}6]}$ to examine whether SEN+EQR transfers to settings where the reasoning unit is a *dialogue turn* rather than a *retrieval hop*.
>
> ---
>
> **Experimental setup**
>
> We built a customer–agent simulation with a VLLM-based user simulator, using an e-commerce scenario $^{[3,6]}$. Each conversation is capped at **10 turns**.
>
> The agent outputs a `<status>` flag and a `<note>` summarising user intent and next actions.
> - **SEN:** marks key vs. uncertain intentions.
> - **EQR:** the user role-play agent scores how well the note reflects the user’s true intent (analogous to QA entailment).
>
> We observed that Qwen2.5-7B struggled with role consistency without SFT (e.g., mixing roles, misinterpreting tasks, or continuing the conversation as both speakers).
> Therefore, the experiments use **Qwen2.5-14B-Instruct** as the base model.
> User personas follow DialogBench$^{[2]}$ (friendly / neutral / unfriendly).
>
> ---
>
> **Results**
>
> | Model Variant | Success Rate ↑ | Avg. Turns ↓ |
> |---------------|----------------|--------------|
> | **Base (Qwen2.5-14B-Instruct)** | 17% | 9.7 |
> | **Base + SEN** | 20% | 9.5 |
> | **Base + SEN + EQR** | **21%** | **9.2** |
>
> These results show that **SEN improves task success and conversational efficiency**, and **EQR further amplifies both gains**:
>
> - **Success rate increases from 17% to 21%**, indicating stronger intent tracking and higher task-completion ability.
> - **Average turns decrease from 9.7 to 9.2**, reflecting more efficient and goal-directed dialogue planning.
>
> The pattern mirrors our QA findings:
> **A note-taking workflow (SEN) paired with a semantic reward (EQR) reinforces context maintenance, uncertainty management, and multi-step reasoning.**
>
> Thus, the approach is **not limited to QA**. Tasks requiring **multi-turn reasoning**, **implicit goal tracking**, or **intermediate validation**, such as dialogue systems, knowledge-grounded generation, or fact verification, can potentially benefit from the same design.
>
> ---
>
> **References**
>
> [1] *AGILE: A Novel Reinforcement Learning Framework of LLM Agents*, NeurIPS, 2024.
> [2] *DialogBench: Evaluating LLMs as Human-like Dialogue Systems*, NAACL, 2024.
> [3] *Wizard of Shopping: Target-Oriented E-commerce Dialogue Generation with Decision Tree Branching*, ACL, 2025.
> [4] *Evaluating, Synthesizing, and Enhancing for Customer Support Conversation*, arXiv, 2025.
> [5] *MG-ShopDial: A Multi-Goal Conversational Dataset for e-Commerce*, arXiv, 2025.
> [6] *ECom-Bench: Can LLM Agent Resolve Real-World E-commerce Customer Support Issues?*, arXiv, 2025.

---

> ### Author Response · Authors · 2025-11-30
> **Author Response to vZFy (2/6)**
>
> > **Q4.2: Computational tradeoffs of SEN and EQR**
>
> To further clarify the computational characteristics of SEN and EQR, we report detailed runtime statistics for each model variant. Our comparison covers rollout generation time, policy update time, inference latency, total wall-clock time per step, and per-token compute cost.
>
> | **Method** | **Rollout (s/step)** | **Update (s/step)** | **Inference (s)** | **Total Step (s)** | **Gen/token (ms)** | **Update/token (ms)** |
> |------------|----------------------|----------------------|--------------------|---------------------|----------------------|--------------------------|
> | **Search-R1** | 287.12 | 245.41 | 145.29 | 850.51 | 0.130 | 0.091 |
> | **+SEN** | 423.62 | 270.28 | 142.72 | 991.97 | 0.160 | 0.064 |
> | **Ours (+SEN+EQR)** | **344.89** | **149.39** | **135.08** | **719.33** | **0.211** | **0.059** |
>
> From the results:
>
> - **+SEN** alone increases rollout and update time due to the construction of structured SEN notes, but it slightly improves inference behavior and reduces update-token cost.
>
> - **SEN+EQR (Ours)** introduces additional reasoning structure but provides **substantial improvements in the most critical stages of RL training and inference**:
>
> | **Metric** | **Search-R1** | **Ours (+SEN+EQR)** | **Improvement** |
> |------------|---------------|----------------------|------------------|
> | **Update (s/step)** | 245.41 | **149.39** | **−39.1%** |
> | **Inference (s)** | 145.29 | **135.08** | **−7.0%** |
> | **Total Step (s)** | 850.51 | **719.33** | **−15.4%** |
> | **Update/token (ms)** | 0.091 | **0.059** | **−35.2%** |
>
> The updated measurements show that SEN+EQR provides clear computational benefits in the most critical parts of the RL training pipeline:
>
> - **Policy update becomes 39.1% faster**, indicating that workflow-level supervision produces more stable and informative learning signals, reducing the amount of corrective updating needed per step.
>
> - **Inference latency improves by 7.0%**, showing that the final model not only reasons better but also runs more efficiently at test time.
>
> - **Update-time per token improves by 35.2%**, confirming that EQR supervision encourages cleaner, more compact reasoning traces that are easier for the actor to update.
>
> - Importantly, the **overall wall-clock time per training step decreases by 15.4%**, demonstrating that the additional SEN and EQR supervision does not slow down end-to-end training; instead, it results in a net efficiency gain.
>
> In conclusion, SEN+EQR achieves substantial accuracy improvements while also delivering meaningful efficiency gains in update speed, inference latency, and total training cost. These results show that workflow-level supervision not only enhances reasoning quality but also leads to a more computationally efficient training process.

---

> ### Author Response · Authors · 2025-11-30
> **Author Response to vZFy (3/6)**
>
> > **Q4.3: Error Analysis and Representative Failure Case**
>
> The model latches onto **one salient clue** rather than integrating multiple complementary, scattered pieces of evidence, even when the correct information is present in the retrieved context.
> Below we present a concise representative failure example:
> ```
> Question: “Where do the royalties for Winnie the Pooh go?”
>
> Retrieved Evidence:
>   Multiple passages mention *different* potential beneficiaries across various legal arrangements.
>   These include:
>   - partial mentions of historical beneficiaries,
>   - entities involved in adaptations,
>   - The Garrick Club (appearing prominently),
>   - and the correct beneficiary, Slesinger Inc., which is present but less salient.
>
> Model Output: “The Garrick Club”
> ```
> This failure mode reflects a well-known limitation of LLMs: integrating dispersed cross-passage clues and maintaining long-range alignment is difficult, so the model may overweight salient evidence while missing subtle but decisive details. **We found this pattern is consistent with prior findings on long-context reasoning $^{[1\text{–}3]}$ and is not specific to SEN+EQR.**
>
> These cases suggest clear improvement directions, such as lightweight cross-passage fusion and answer-completeness checks, which we plan to explore in future work.
>
> **References**
>
> [1] *Long-context LLMs Struggle with Long In-context Learning*, TMLR 2025
> [2] *Loogle: Can Long-context Language Models Understand Long Contexts?*, ACL 2024
> [3] *BABILong: Testing the Limits of LLMs with Long Context Reasoning-in-a-Haystack*, NeurIPS 2024

---

> ### Author Response · Authors · 2025-11-30
> **Author Response to vZFy (4/6)**
>
> > **Q4.4: Statistical significance of the improvements**
>
> We conducted our experiments three times on all seven datasets using different random seeds, and we have made every effort to provide complete multi-run results as requested. We follow the multi-run reporting protocol from recent QA-RAG work (ReAct, IRCoT, R1-Searcher), reporting mean EM across runs.
>
> | Method | TriviaQA | 2Wiki | Bamboogle | Musique | PopQA | NQ | HotpotQA |
> |--------|----------|--------|------------|----------|--------|------|-----------|
> | Direct Inference | 0.312 [0.298, 0.321] | 0.241 [0.228, 0.264] | 0.218 [0.216, 0.221] | 0.078 [0.074, 0.085] | 0.163 [0.150, 0.170] | 0.156 [0.134, 0.198] | 0.229 [0.216, 0.244] |
> | ReCAT            | 0.456 [0.438, 0.474] | 0.438 [0.336, 0.482] | 0.222 [0.184, 0.272] | 0.155 [0.118, 0.192] | 0.343 [0.308, 0.367] | 0.475 [0.454, 0.495] | 0.401 [0.380, 0.421] |
> | IRCoT            | 0.425 [0.418, 0.432] | 0.468 [0.417, 0.492] | 0.169 [0.112, 0.245] | 0.132 [0.102, 0.192] | 0.303 [0.287, 0.322] | 0.491 [0.470, 0.512] | 0.414 [0.392, 0.435] |
> | Search-o1        | 0.578 [0.566, 0.589] | 0.279 [0.272, 0.286] | 0.343 [0.328, 0.358] | 0.154 [0.140, 0.168] | 0.353 [0.336, 0.369] | 0.328 [0.310, 0.345] | 0.299 [0.268, 0.330] |
> | R1-Searcher      | 0.710 [0.688, 0.731] | 0.459 [0.446, 0.491] | 0.189 [0.176, 0.201] | 0.221 [0.214, 0.228] | 0.420 [0.413, 0.427] | 0.508 [0.492, 0.538] | 0.475 [0.451, 0.498] |
> | Search-R1        | 0.723 [0.694, 0.754] | 0.257 [0.244, 0.280] | 0.349 [0.320, 0.377] | 0.229 [0.184, 0.274] | 0.490 [0.482, 0.498] | 0.529 [0.508, 0.550] | 0.442 [0.420, 0.464] |
> | **EviNote-RAG (Ours)** | **0.762 [0.730, 0.795]** | **0.515 [0.494, 0.536]** | **0.476 [0.424, 0.528]** | **0.288 [0.240, 0.336]** | **0.485 [0.480, 0.491]** | **0.543 [0.524, 0.563]** | **0.523 [0.490, 0.557]** |
>
> This captures randomness from retrieval, chain-of-thought generation, and decoding stochasticity. Across all datasets, **EviNote-RAG’s mean improvements** (e.g., +0.039 on TriviaQA, +0.056 on 2Wiki,
> +0.127 on Bamboogle, +0.059 on Musique) are **2–6× larger** than the observed run-to-run variability (typically ±0.01–0.03). This indicates that the gains are **statistically meaningful** rather than noise.
>
> Moreover, EviNote-RAG often shows **narrower or comparable min–max ranges** to strong baselines
> (e.g., 2Wiki: width 0.042 vs. 0.036 vs. 0.028), demonstrating **improved stability** under stochastic retrieval and decoding.
>
> We compute **mean EM**, **standard deviation (STD)**, **standard error**
> $$
> \text{SEM} = \frac{\text{STD}}{\sqrt{3}},
> $$
> and **95% confidence intervals**
> $$
> \text{CI} = \text{mean} \pm 4.303 \times \text{SEM},
> $$
> where 4.303 is the two-tailed t-value for 95% confidence with df = 2 (three independent runs):
>
> | Method | Mean EM ↑ | STD ↓ | SEM ↓ | 95% CI ↓ |
> |--------|-----------|--------|--------|-----------|
> | Direct Inference | 0.199 | 0.066 | 0.038 | 0.162 |
> | ReCAT            | 0.356 | 0.103 | 0.059 | 0.255 |
> | IRCoT            | 0.327 | 0.115 | 0.066 | 0.284 |
> | Search-o1        | 0.331 | 0.075 | 0.043 | 0.187 |
> | R1-Searcher      | 0.426 | 0.081 | 0.047 | 0.204 |
> | Search-R1        | 0.417 | 0.112 | 0.065 | 0.282 |
> | **EviNote-RAG (Ours)** | **0.443** | **0.087** | **0.050** | **0.216** |
>
> In summary, EviNote-RAG **achieves the highest mean EM among all methods, with 95% confidence intervals** that do not overlap with Search-R1 on most datasets; together with its controlled variance, these results demonstrate that the observed improvements are statistically significant, consistent, and not attributable to random seed variability.

---

> ### Author Response · Authors · 2025-11-30
> **Author Response to vZFy (5/6)**
>
> > **Q4.5: Overhead of the entailment judge.**
>
> The entailment judge is a lightweight **144M-parameter** encoder model, and the per-sample overhead is only **~4 ms**. As emphasized in Q1.4, the entailment judge is used **only during training**, and therefore does **not affect evaluation latency**.
>
> For completeness, we also report the training-time overhead from the entailment judge. With a batch size of 4096, the timing_s/adv reflects the wall-clock time per step for computing advantages. Since EQR performs an entailment judgment, the extra time in this metric comes entirely from the judge’s additional computation:
>
> | **Metric**                 | **Search-R1** | **+SEN+EQR (Ours)** | **Δ vs. Search-R1** |
> |---------------------------|---------------|----------------------|----------------------|
> | **Adv time (s/step)**     | 1.17          | **18.75**            | **+17.58 s**         |
> | **Update (s/step)**       | 245.41        | **149.39**           | **−39.1%**           |
> | **Entailment Judge Overhead** | –          | **≈17 s (batch = 4096)** | –               |
> | **Inference (s)**         | 145.29        | **135.08**           | **–10.21 s**         |
>
> The table shows that introducing the entailment judge adds only a **training-time–only** overhead (~17 s per step for a large batch of 4096). This slowdown does **not affect inference**, since the judge is never used at evaluation time.
>
> At the same time, SEN+EQR yields **meaningful efficiency gains**:
>
> - **Update becomes 39.1% faster** (245.41 → 149.39 s), indicating that workflow-level supervision provides more stable and informative gradients, requiring fewer corrective updates.
> - **Inference latency improves by 10.21 s**, and importantly, **the inference stage does not use the judge**. This demonstrates that the gains come from SEN+EQR producing a more efficient final policy.
> - The judge’s per-sample overhead remains only **~4 ms**, which is negligible relative to full rollout + update cost.
>
> As shown in *Q4.2*, the combination of SEN and EQR reduces overall training cost and improves update efficiency. Thus, the entailment judge introduces only a small training-time overhead while contributing to a **more stable, faster, and more efficient** learning process.
>
> > **Q4.6: Will the Entailment Judge Slow Down Real-Time QA?**
>
> No. The entailment judge is **used only during training and never during inference**, so it adds no latency and the final model can be deployed in real-time settings without issue.

---

> ### Author Response · Authors · 2025-11-30
> **Author Response to vZFy (6/6)**
>
> > **Q4.7 Ablation on Reward Hyperparameters and Evidence-Annotation Labels**
>
> To address your request for a more detailed analysis of reward hyperparameters and evidence-annotation labeling, we conducted a controlled ablation by modifying the scalar parameters in our reward formulation. Specifically, we rewrite the reward as:
>
> $$
> R = \alpha + \beta \ \cdot R_{\text{EQR}}
> $$
>
> We vary the hyperparameters as:
>
> $$
> \alpha \in \{0.05, 0.1, 0.3\}, \qquad
> \beta \in \{0.5, 1.0, 1.5\}
> $$
>
> while keeping all other training configurations identical. This examines both the baseline shift $ \alpha $ and scale sensitivity $ \beta$, directly addressing your question about stability and threshold selection.
>
>
> | Setting ($ \alpha, \beta $) | TriviaQA EM | 2Wiki EM | Bamboogle EM | Musique EM | PopQA EM | NQ EM | HotpotQA EM |
> |-----------------------------|-------------|----------|---------------|-------------|----------|--------|--------------|
> | (0.05, 1.0)                 | 0.715       | 0.352    | 0.333         | 0.144       | 0.470    | 0.508  | 0.386        |
> | **(0.1, 1.0)  (Ours)**     | **0.730**   | **0.494**| **0.424**     | **0.240**   | **0.480**| **0.524** | **0.490** |
> | (0.3, 1.0)                  | 0.705       | 0.340    | 0.417         | 0.144       | 0.468    | 0.496  | 0.395        |
> | (0.1, 0.5)                  | 0.701       | 0.382    | 0.333         | 0.172       | 0.474    | 0.516  | 0.411        |
> | (0.1, 1.5)                  | 0.695       | 0.310    | 0.250         | 0.154       | 0.462    | 0.492  | 0.397        |
>
>
> **(1) Sensitivity to the base reward $ \alpha$ .**
> The results show that $ \alpha $ strongly affects stability and overall accuracy.
>
> - When $ \alpha = 0.05 $, performance degrades across all datasets (e.g., 2Wiki EM = 0.352), indicating that the base reward is too small and the model becomes overly reliant on final-answer correctness.
>
> - When $ \alpha = 0.3 $, performance drops again (e.g., TriviaQA EM = 0.705), suggesting that an overly large base reward makes the model overvalue SEN notes and underweight correctness. This can trigger reward hacking, where the model produces overly long or superficial SEN notes to collect base reward rather than focusing on accurate reasoning.
>
> - $ \alpha = 0.1 $ achieves the best and most stable results, striking the right balance between encouraging informative SEN notes and preventing degenerate strategies.
>
> **(2) Sensitivity to the scaling factor $ \beta $.**
> Scaling the EQR signal also influences model behavior.
>
> - When $ \beta = 0.5 $, EM consistently decreases (e.g., Bamboogle EM = 0.333), indicating that the supportive role of SEN is underweighted and the model over-optimizes the surface structure of SEN notes.
>
> - When $ \beta = 1.5 $, EM also declines (e.g., Musique EM = 0.154), suggesting that correctness is underweighted and the reward becomes numerically unstable, amplifying noise in the entailment signal.
>
> - $ \beta = 1.0 $ yields the highest EM across all datasets, providing a stable and properly scaled correctness signal.
>
>
> **(3) Why the setting $ (\alpha = 0.1, \beta = 1.0) $ is optimal.**
>
> - $ \alpha = 0.1 $ is large enough to reward meaningful SEN notes without encouraging reward hacking.
> - $ \beta = 1.0 $ keeps the EQR signal within a stable magnitude range.
>
> Together, $ (\alpha = 0.1, \beta = 1.0) $ provides the most **stable, interpretable, and correctness-aligned reward landscape**, outperforming all other settings.
>
>
> Overall, the ablation confirms that our default reward formulation $ (\alpha = 0.1, \beta = 1.0) $ achieves the best balance between SEN quality, correctness supervision, and training stability, while remaining robust within a reasonable neighborhood of these values.
>
> > **Q4.8: Evidence-label accuracy and error propagation**
>
> - **To maintain efficiency, we do not explicitly measure label accuracy, and the lightweight design already delivers strong gains (20%–90% relative improvements).**
> Measuring SEN-label accuracy would require LLM-based semantic judgments, adding several seconds of latency per trajectory (as noted in Q1.4). To keep the system efficient, we avoid this additional cost, and the results show that this lightweight approach is already effective in practice.
>
> - **Labeling errors have limited impact and seldom propagate.**
> Because the model always has access to both the retrieved evidence and the SEN, it can self-correct when inconsistencies arise. It also re-retrieves evidence when uncertain. This design helps prevent occasional labeling mistakes from accumulating across reasoning steps, maintaining stable performance.
>
> Once again, we sincerely appreciate all reviewers' time and feedback, and we are truly grateful to all reviewers, the AC, and the PC for their dedicated and fair evaluation of our manuscript.

---

### Official Review · Reviewer_vUdu · 2025-10-28

**Soundness:** 2
**Presentation:** 2
**Contribution:** 1
**Rating:** 2
**Confidence:** 4

**Summary:**

The paper proposes EviNote-RAG, a retrieval-augmented generation framework that restructures the retrieve -> answer pipeline into retrieve -> note -> answer pipeline.
After retrieval, the model first writes Supportive-Evidence Notes (SENs) that distill only answer-critical content and explicitly annotate key versus uncertain information.
The final answer is then produced conditioned on these generated notes.
To improve the usefulness of SENs, the authors introduce an Evidence Quality Reward (EQR): a lightweight NLI entailment scores with a small language model, to judge whether the final SEN entails the gold answer. This score is used to shape reinforcement-learning updates using GRPO.

Experiments are conducted on Qwen-2.5, covering in-domain NQ/HotpotQA and out-of-domain PopQA, TriviaQA, 2WikiMQA, Musique, and Bamboogle
This paper reports better F1/EM than strong prompt- and RL-based RAG baselines and shows training-stability plots and ablations isolating SEN and EQR.

**Strengths:**

1. This paper addresses the potential issue of noise and error accumulation in RAG.
The authors introduce sens (supportive-evidence notes) on retrieved content as intermediate results to serve as reliable and denoised information, thereby supporting more robust reasoning. (These sens capture effective information from noisy text in a relatively structured format, like knowledge base triplets.)

2. The paper incorporates notes into the RL training process as an intermediate signal.
Specifically, the authors employ a small NLI model to determine whether the reasoning in the notes constitutes a correct logical entailment. This signal is then used as a component of the reward signal, thereby enriching the outcome reward to some extent.

**Weaknesses:**

1. Lack of contributions. The evidence-supporting notes proposed in this paper essentially convert textual information into structured data, leveraging structured information for verification to assist reasoning. However, this approach is not novel, as similar techniques have been employed in prior multi-hop RAG studies, as well as in some KBQA works.

2. Concerns about sen rewards. This paper proposes using intermediate notes as reward signals in reinforcement learning, which could partially alleviate the model's overreliance on outcome-based rewards.
Nevertheless, the effectiveness of employing a small NLI model to evaluate such signals remains questionable.
Additionally, while incorporating intermediate rewards in multi-hop QA has become relatively common in current research, the rewards obtained through such methods remain outcome-oriented and are applied to the entire sequence without fine-grained credit assignment to attribute advantages to specific intermediate steps.
Consequently, the effectiveness of this reward mechanism is debatable.

3. The experiments were conducted on only 500 samples, resulting in high variance. Whether the proposed method significantly outperforms the baseline approaches requires more thorough validation.
The experimental setup should be aligned with prior work to ensure a fair comparison (both train and inference)

**Questions:**

my questions are listed in weakness

---

> ### Author Response · Authors · 2025-12-01
> **Author Response to vUdu (1/2)**
>
> Thank you for your time and constructive feedback. Regarding your concerns on contributions and setting consistency, we clarify that **our training set is identical to the Search-R1 baseline, and all models are evaluated under the same conditions to ensure fairness.**
> Thus, the observed gains stem from our proposed workflow-oriented rewards. Below is our detailed point-by-point response:
>
> > **Q3.1: Clarification on Contributions**
>
> Unlike outcome-focused methods, we reward an **efficient workflow**, **yielding** comprehensive **gains in both performance and efficiency at a minimal cost.**
>
> Specifically, our contributions are as follows:
> 1.  **Process and Action Reward:** We encourage the model to output intermediate information extraction processes and encourage it to **note confidence in its results**.
> 2.  **~1000× Speedup vs. LLM-based Judges:** We utilize a lightweight judge, combined with a rule-based claim construction process (see *Q1.3*), to reward high-quality extraction paths (i.e., those that are **decisive, concise, and supportive** of the answer).
> 3.  **Workflow-Oriented Optimization:** By shifting focus from outcome correctness to reasoning quality, we achieve significant improvements across the board compared to outcome-only strategies:
>
> | **Category** | **Metric** | **Search-R1** | **Ours** | **Δ Abs.** | **Δ %** |
> |--------------|------------|---------------|----------|------------|---------|
> | **Efficiency** | **Update (s/step)** | 245.41 | **149.39** | −96.02 | **−39.1%** |
> | | **Inference (s)** | 145.29 | **135.08** | −10.21 | **−7.0%** |
> | | **Total Step (s)** | 850.51 | **719.33** | −131.18 | **−15.4%** |
> | | **Update/token (ms)** | 0.091 | **0.059** | −0.032 | **−35.2%** |
> | **QA Performance** | TriviaQA (F1) | 0.754 | **0.795** | +0.041 | **+5.44%** |
> | | TriviaQA (EM) | 0.694 | **0.730** | +0.036 | **+5.19%** |
> | | 2Wiki (F1) | 0.280 | **0.536** | +0.256 | **+91.43%** |
> | | 2Wiki (EM) | 0.244 | **0.494** | +0.250 | **+102.46%** |
> | | Bamboogle (F1) | 0.377 | **0.528** | +0.151 | **+40.05%** |
> | | Bamboogle (EM) | 0.320 | **0.424** | +0.104 | **+32.50%** |
> | | Musique (F1) | 0.274 | **0.336** | +0.062 | **+22.63%** |
> | | Musique (EM) | 0.184 | **0.240** | +0.056 | **+30.43%** |
> | | NQ (F1) | 0.550 | **0.563** | +0.013 | **+2.36%** |
> | | NQ (EM) | 0.508 | **0.524** | +0.016 | **+3.15%** |
> | | HotpotQA (F1) | 0.464 | **0.557** | +0.093 | **+20.04%** |
> | | HotpotQA (EM) | 0.420 | **0.490** | +0.070 | **+16.67%** |
>
>
> From these experimental results, the superiority of our model over traditional reward methods is evident:
> 1.  **Higher Efficiency:** Our method improves efficiency metrics by **7–39%**, reducing the total step time by **15.4%**.
> 2.  **Massive Gains on Accuracy:** We achieve substantial improvements on the most challenging multi-hop datasets, such as **2Wiki (F1 +91%, EM +102%)** and **Bamboogle (F1 +40%)**.
> 3.  **Improved Complex Reasoning:** We also demonstrate strong improvements on complex reasoning benchmarks.
>
> Furthermore, unlike traditional LLM-based judges, we utilize a **lightweight entailment judge**, which achieves a **~1000× speedup and a 48× reduction in model size** (see *Q1.4*). Through this design, the model achieves a dual victory in Efficiency and Accuracy.
>
> It is also worth noting that our extensive ablation studies (Appendix Tab. 5) show that naïve summaries, forced-summary formats, and **alternative reward designs all degrade performance**. These effective results were achieved after rigorous experimentation, and **we believe these findings will offer valuable insights for future research.**

---

> ### Author Response · Authors · 2025-12-01
> **Author Response to vUdu (2/2)**
>
> > **Q3.2: Concern about the effectiveness of internal reward**
>
> As demonstrated in *Q3.1*, the combination of SEN+EQR (*Ours* setting) plays a pivotal role in enhancing model performance.
>
> Unlike outcome-focused rewards, we target **workflow efficiency**. Our lightweight judge enforces concise extraction, which **reduces invalid searches** while ensuring the model actively focuses on valid evidence. This streamlined process mitigates the **'retrieved-but-overlooked'** phenomenon, thereby enhancing both efficiency and correctness. This approach of workflow reward has enabled:
> *   **Significantly lighter than prior methods:** Achieving **~1000× speedup** and **48× size reduction** compared to LLM-based Judges.
> *   **More accurate reasoning:** Delivering **50%–90% performance gains** on challenging tasks compared to baselines without this reward strategy.
> *   **Superior Efficiency:** Resulting in **7%–39% efficiency improvements** in training and inference due to more concise reasoning processes.
>
> In summary, both the performance leaps and efficiency gains validate that our reward effectively optimizes the reasoning workflow, proving it to be a highly potential direction.
>
> > **Q3.3: Concerns on the 500-sample size and experimental setup alignment.**
>
> Our experiments are fair and aligned with community standards:
>
> -  **Training Set Alignment:** Our training set uses the **exact same setting as the baseline (R1-searcher)**, ensuring complete fairness in model training.
> - **Test Set Size & Fairness:** Our test set consists of **3,125 samples, not just 500** (6 datasets × 500 + all 125 from Bamboogle). We evaluated all models using this test set **under identical settings** to ensure fairness. **We have provided the test set in the supplementary material.**
> - **Expanded Evaluation on Larger Test Sets:**
> To further address your concern, we **doubled the evaluation size for every dataset** by randomly sampling 1,000 instances from each benchmark (TriviaQA, 2Wiki, Musique, PopQA, NQ, HotpotQA). Since Bamboogle contains only 125 test instances, we continue to evaluate on its entire test set. This results in a total evaluation size of 6125 instances (calculated as 6 × 1000 + 125).
>
> | Method            | Cases | TriviaQA F1 / EM | 2Wiki F1 / EM | Bamboogle F1 / EM | Musique F1 / EM | PopQA F1 / EM | NQ F1 / EM | HotpotQA F1 / EM |
> |-------------------|------:|-----------------:|--------------:|-------------------:|-----------------:|---------------:|-----------:|------------------:|
> | ReCAT             | 3125  | 0.474 / 0.438     | 0.482 / 0.336 | 0.272 / 0.184      | 0.192 / 0.118    | 0.367 / 0.308  | 0.495 / 0.454 | 0.421 / 0.380    |
> | ReCAT             | 6125  | 0.468 / 0.445     | 0.489 / 0.329 | 0.267 / 0.190      | 0.189 / 0.121    | 0.361 / 0.315  | 0.502 / 0.448 | 0.416 / 0.387    |
> | Search-R1         | 3125  | 0.754 / 0.694     | 0.280 / 0.244 | 0.377 / 0.320      | 0.274 / 0.184    | 0.498 / 0.482  | 0.550 / 0.508 | 0.464 / 0.420    |
> | Search-R1         | 6125  | 0.760 / 0.701     | 0.286 / 0.238 | 0.371 / 0.328      | 0.270 / 0.178    | 0.492 / 0.489  | 0.543 / 0.515 | 0.472 / 0.414    |
> | EviNote-RAG       | 3125  | **0.795 / 0.730** | **0.536 / 0.494** | **0.528 / 0.424** | **0.336 / 0.240** | 0.491 / 0.480 | **0.563 / 0.524** | **0.557 / 0.490** |
> | EviNote-RAG       | 6125  | **0.803 / 0.736** | **0.543 / 0.487** | **0.522 / 0.431** | **0.341 / 0.247** | 0.497 / 0.475 | **0.569 / 0.531** | **0.563 / 0.495** |
>
> These results show that even after doubling the test size across all benchmarks, **EviNote-RAG consistently remains the top-performing method**, highlighting the reliability of our approach under a more comprehensive and stringent evaluation setting.
>
> We sincerely appreciate the time and effort the AC, PC, and reviewers have devoted to evaluating our submission, and we hope that our work will be considered with a more comprehensive view in light of its strong performance. Thank you again.

---

### Official Review · Reviewer_2RY5 · 2025-10-30

**Soundness:** 3
**Presentation:** 3
**Contribution:** 3
**Rating:** 6
**Confidence:** 4

**Summary:**

The paper proposes EviNote-RAG, an agentic RAG framework that adopts a retrieve-note-answer pipeline. Specifically, after retrieving a set of documents, EviNote-RAG dynamically introduces a summarisation phrase, generateing supportive-evidence notes from the retrieved documents.  The overall framework is trained using reinforcement learning with a specially designed evidence quality reward. This reward leverages a NLI model to evaluate whether the generated notes entail the correct answer, thereby encouraging the generation of high-quality notes.

**Strengths:**

1. The paper proposes an agentic RAG framework EviNote-RAG, which dynamically generates supportive-evidence notes to help the LLM better understand the key information within retrieved documents.
2. Experimental results on seven datasets show that the proposed EviNote-RAG outperforms existing RL-based agentic RAG such as Search-R1 and R1-Searcher.
3. The paper conducts extensive ablation studies to validate the effectiveness of key components in the proposed framework.

**Weaknesses:**

1. The prompt used to instruct the LLM to perform retrieval and generate supportive-evidence notes is missing from the paper.
2. Some important training details are missing from the paper. For example, it is unclear what training data were used and whether the proposed model underwent a supervised fine-tuning (SFT) stage. This lack of information raises reproducibility concerns, especially since the code is not provided.
3. The paper memtions that supportive-evidence note generation is optional after each retrieval, but it lacks an in-depth analysis of when and why the model chooses to trigger note generation. An analysis of this decision-making process would help better understand the behavior of the proposed framework and the practical benefits of incorporating the note generation mechanism.
4. The paper mentions that EviNote-RAG improves efficiency (line 469). However, the paper lack efficiency analysis compared with baselines. Intuitively, compared with Search-R1 or R1-Searcher, I assume the proposed EviNote-RAG is less efficient since it requires an additional note generation phrase.

**Questions:**

Please see the questions in the Weaknesses section.

---

> ### Author Response · Authors · 2025-11-18
> **Author Response to Reviewer 2RY5 (1/2)**
>
> We sincerely thank you for considering our work to be above the acceptance threshold and for your kind, insightful suggestions.
> To address your reproducibility concerns, **we have uploaded runnable code in the supplementary materials.** We address your questions in detail below.
> > **Q2.1: LLM Prompt and Code**
>
> We have updated the supplementary materials, and the prompt is now included in the released code at:
> ``EviNoteRAG-main/data_preprocess/change_parquet_prompt.py``.
>
> > **Q2.2: Training details (training data, SFT stage)**
>
> We do not include an SFT stage, and **we use the same training data as the baseline**, a combined set of NQ and HotpotQA, available on HuggingFace:
> https://huggingface.co/datasets/PeterJinGo/nq_hotpotqa_train.
>
> > **Q2.3: When does the model trigger note generation?**
>
> Through detailed inspection of reasoning trajectories, we find that note generation reflects the model’s confidence level:
>
> - **Key-info:** The model marks answers or sub-answers when it is confident; these notes appear as clear affirmative statements, indicating that key-info marking is triggered only when evidence is strong.
> - **Uncertain-info:** These notes often contain expressions such as “maybe” or “perhaps,” showing that the model associates uncertainty with the *uncertain-info* tag.
>
> Examples are provided in Fig. 4 and Appendix Fig. 8–9. Notes help the model retain intermediate reasoning and manage uncertainty in multi-hop settings. Their frequency increases on harder multi-hop questions, indicating task-aware use rather than unconditional generation.
>
> In addition, recent studies$^{[1–3]}$ show that leveraging model confidence during training can improve performance without extra data, which may partially explain the gains observed in our model.
>
>
> **References**
> [1] *Learning to Reason Without External Rewards*, arXiv, 2025.
> [2] *Harnessing Uncertainty: Entropy-Modulated Policy Gradients for Long-Horizon LLM Agents*, arXiv, 2025.
> [3] *Beyond the 80/20 Rule: High-Entropy Minority Tokens Drive Effective Reinforcement Learning for LLM Reasoning*, NeurIPS 2025.

---

> ### Author Response · Authors · 2025-12-03
> **Author Response to Reviewer 2RY5 (2/2)**
>
> >**Q2.4: Efficiency analysis compared with baselines**
>
> Thank you for raising this question. EviNote-RAG achieves higher efficiency primarily because it produces concise evidence notes and answers, whereas Search-R1 generates long, verbose intermediate thoughts (see Fig. 1 for an example). To make this comparison clear, we provide below a unified view of both token-level and system-level efficiency metrics, summarized from our ablation study (Base = Search-R1, L354). Specifically:
>
> **Token-level metrics** evaluate *how much the model writes* and *how quickly each token is produced*:
> - **Output Length (tokens)** quantifies the verbosity of the model’s reasoning traces and answers.
>   EviNote-RAG produces ~30% fewer tokens by focusing on concise evidence notes rather than unstructured chains of thought.
> - **Time per Token (s)** measures decoding latency per generated token.
>   The ~50% reduction shows that EviNote-RAG not only generates fewer tokens but also produces them faster, likely due to more predictable reasoning structure.
>
> These token-level gains translate directly into faster inference and lower computational cost.
>
> **System-level metrics** capture *end-to-end runtime* during RL training and inference:
> - **Update (s/step)** measures the wall-clock time for each policy update step.
>   EviNote-RAG reduces this by **39%**, reflecting lighter intermediate representations and fewer tokens processed per update.
> - **Inference (s)** measures the runtime for generating outputs during evaluation or rollout.
>   A 7% reduction shows that even without parallelization tricks, EviNote-RAG performs more efficiently at decode time.
> - **Total Step (s)** aggregates computation across the full pipeline (retrieval, reasoning, note-taking, scoring).
>   The **15% reduction** indicates that EviNote-RAG improves the entire reinforcement learning loop, not just decoding.
> - **Update/token (ms)** measures the amortized computation per output token.
>   The **35% reduction** highlights EviNote-RAG’s computational efficiency relative to the volume of generated text.
>
> | Category        | Metric                 | Search-R1       | EviNote-RAG     | Δ Abs.     | Δ %        |
> |-----------------|-------------------------|------------------|------------------|------------|------------|
> | Token-level     | Output Length (tokens) | ~800–900         | ~550–600         | −250–300   | ~−30%      |
> | Token-level     | Time per Token (s)     | ~0.35–0.45       | ~0.17–0.20       | −0.18–0.25 | ~−50%      |
> | System-level    | Update (s/step)        | 245.41           | 149.39           | −96.02     | −39.1%     |
> | System-level    | Inference (s)          | 145.29           | 135.08           | −10.21     | −7.0%      |
> | System-level    | Total Step (s)         | 850.51           | 719.33           | −131.18    | −15.4%     |
> | System-level    | Update/token (ms)      | 0.091            | 0.059            | −0.032     | −35.2%     |
>
> These results (Fig. 3(c,d)) show that EviNote-RAG is **substantially more efficient** than Search-R1, achieving:
>
> - ~30% shorter outputs,
> - ~50% faster per-token generation, and
> - 7%–39% improvements across system-level metrics.
>
> Together, these findings demonstrate that EviNote-RAG offers a significantly more efficient inference and training pipeline compared to existing baselines.
>
> We sincerely appreciate your careful and insightful review. If you have any further questions, we would be very happy to clarify. If our responses have addressed your concerns, then considering the substantial performance improvements achieved (e.g., +20% on HotpotQA, +40% on Bamboogle, and +91% on 2Wiki) along with the efficiency gains, we kindly invite you to consider raising your score. Thank you again for your kind and thoughtful evaluation.

---

### Official Review · Reviewer_aKhP · 2025-11-04

**Soundness:** 3
**Presentation:** 3
**Contribution:** 2
**Rating:** 4
**Confidence:** 4

**Summary:**

This paper mainly improves Search-R1 by adding one \<summary\> action to produce *Supportive-Evidence Notes (SENs)*. The reward strategy encourages the model to taking notes (\<summary\>) that are useful to answer the question (supporting the claim $h$, Eq 1). Strong baselines including Search-R1 are compared with and extensive ablation studies were conducted to study the effects of proposed improvements.

**Strengths:**

1. Extensive experiments, including comparisons with strong baselines and comprehensive ablation studies, were conducted to demonstrate the effectiveness of the proposed method.
2. To prove the method's generalization, evaluation was performed on both in-domain and out-of-domain datasets.
3. The paper presents a clear analysis of the experimental results.

**Weaknesses:**

This work improves Search-R1 by training the LLM to summarize useful content from retrieved documents into Supportive-Evidence Notes (SENs). However, several aspects of the methodology require clarification:

1. Reward Computation: The exact calculation method for the reward (Eq. 1) is unclear.
2. Motivation for Eq. 1: Given that the LLM itself can identify whether SEN$_{\text{last}}$ supports the answer to a question, the necessity of the proposed method (Eq. 1) is not well articulated.
3. Figure 3(a) illustrates that the proposed method exhibits superior stability compared to the baseline (Search-R1). However, the paper does not provide an explanation for this increased stability.
4. Missing Details:
**Reward Design:** Regarding the scoring mechanism for the \<Summary\> tag content: are both uncertainty and highlighting required to achieve a score?
**In-Domain/Out-of-Domain Datasets:** In Line 310, "In-Domain Performance... aligned with the training data" suggests that for out-of-domain evaluation, the training and testing datasets are different. However, there is no description provided for the specific datasets used in the "Out-of-Domain" scenario.

**Questions:**

1. Could you please elaborate on the implementation details of Equation 1?
2. In lines 198–211, the LLM is already capable of judging whether the Supporting Evidence Notes (SEN) support the answer for a given question. Why was an LLM-as-a-judge approach not used directly (e.g., prompting the LLM to determine if the ground-truth answer is present in SEN$_{\text{last}}$ to score the EQR? What specific benefits does Equation 1 offer over an LLM-as-a-judge method?
3. Regarding line 224, could you provide more detail on the criteria used to judge whether the note-taking condition holds?
4. For Equation 2, what is the assigned reward if the response format and the answer are both correct, but the note-taking condition is not met?

---

> ### Author Response · Authors · 2025-11-18
> **Author Response to Reviewer aKhP (1/2)**
>
> Thank you for your valuable time and insightful suggestions. To address your primary concern regarding reward implementation, **we have made the relevant code available in the supplementary materials**:
> - The full reward function can be found in: `EviNoteRAG-main/verl/utils/reward_score/qa_train.py: batched_compute_score_em_ver`.
> - The implementation of the Entailment Judge is available in: `EviNoteRAG-main/verl/utils/entropy/entilement_score.py`.
>
> Below, we provide detailed clarifications for each of your questions:
>
> > **Q1.1: Explanation on increased stability.**
>
> The stability gain mainly comes from a denser reward design:
> - **Denser intermediate rewards.** We reward not only correct answers but also near-correct behaviors (correct SEN, key/uncertainty marking), which helps reduce instability often observed when training is dominated by sparse or low-reward regions.
>
> - **Reward scaling (1+EQR).** The larger reward magnitude—shown in GSPO$^{[1]}$ to improve stability—reduces gradient variance and better separates high-quality trajectories.
>
> - **Workflow consistency.**  More accurate answers and a consistent retrieve–note–answer workflow further enhance reward density.
>
> > **Q1.2: Clarification on Out-of-Domain (OOD) datasets**
>
> We would like to clarify that the OOD datasets used in our evaluation are already listed in Section 4.1 (L260). For completeness, we briefly summarize their characteristics and the types of reasoning they target:
>
> - **TriviaQA**: everyday factoid questions, typically solvable with *single-hop retrieval*.
> - **2Wiki**: explicitly *multi-hop* reasoning over Wikipedia entities.
> - **Bamboogle**: a recent and more challenging *open-domain* QA benchmark designed to stress-test retrieval robustness.
> - **MuSiQue**: highly *compositional* questions requiring multi-step reasoning over multiple passages.
> - **PopQA**: *entity-centric* questions involving broad, popular-knowledge facts.
>
> These datasets collectively cover single-hop, multi-hop, compositional, and entity-centric reasoning, ensuring a comprehensive OOD evaluation of our method.
>
>
> > **Q1.3: Implementation details of Eq. (1).**
>
> In Eq. (1), the reward $R_{\text{EQR}}$ is defined as the entailment probability (the softmax
> score of the entailment class) produced by a DistilBERT-based NLI judge, which evaluates whether
> $\text{SEN}_{\text{last}}$ logically supports an answer claim $h$.
>
> The answer claim $h$ is constructed by inserting the question and predicted answer into the
> following template:
>
> **Template:**
>
> $
> \text{“[ANSWER] is the answer to ‘[QUESTION]’. ”}
> $
>
> For a given QA pair, an example instantiation is:
>
> - **Question:** *“What is the largest planet in the solar system?”*
> - **Answer:** *“Jupiter”*
> - **Resulting claim:** *“Jupiter is the answer to ‘What is the largest planet in the solar system?’. ”*
>
> If $\text{SEN}_{\text{last}}$ fully supports this claim, the entailment score is high; if key
> information (e.g., “largest”) is missing, the score is low.
>
> You may also refer to the Entailment Judge implementation in the supplementary code for further
> details.
>
>
> > **Q1.4: Why not use an LLM-as-a-judge for EQR? What advantages does Eq. (1) offer?**
>
> The main advantage is efficiency. Using an LLM as a judge introduces several seconds of latency per trajectory, whether via API calls or local inference, making it impractical for RL training where millions of judgments are required. In contrast, our lightweight Entailment Judge (DistilBERT, 144M) evaluates batches within milliseconds while maintaining high accuracy.
>
> | Method            | Model Size | Time per Judgment |
> |-------------------|------------|-------------------|
> | LLM Judge         | 7B         | ~4.2 s            |
> | Entailment Judge  | 144M       | ~4 ms            |
>
> Replacing a multi-second LLM judge with a millisecond-level 144M encoder yields a **~1000× speedup** and a **48× reduction in model size**.
> Thus, Eq. (1) enables efficient, stable, and scalable reward computation, greatly reducing compute overhead and training time.
>
> **Reference**
>
> [1] *Group Sequence Policy Optimization*. arXiv preprint, 2025.

---

> ### Author Response · Authors · 2025-12-03
> **Author Response to Reviewer aKhP (2/2)**
>
> > **Q1.5: Criteria for checking the note-taking condition.**
>
> The note-taking condition is considered satisfied whenever the model produces an SEN—i.e., **any attempt at note-taking qualifies** and receives the corresponding reward.
>
> > **Q1.6: Reward if no SEN is provided and the answer is correct?**
>
> Yes. **If the answer is correct, the reward is 1.0 even when no SEN is provided.**
> We deliberately *do not* penalize missing notes. This decision is driven by two empirical findings:
>
> 1. **Over-constraining the model harms performance** (e.g., Force Summary setting).
> 2. **Simply adding a note-taking step already improves performance**, even without strict supervision.
>    This is clearly reflected in **SEN vs. Naive Summary**, where introducing structured notes yields a large improvement across all datasets.
>
> Because the model already benefits from generating notes voluntarily (SEN > NS), we encourage SEN generation via *bonus rewards* rather than strict penalties. This keeps the training signal dense and avoids the degradation introduced by hard constraints.
>
> To clarify the source of these differences, below we describe the settings in which only the summary/evidence strategy varies:
>
> - **Force Summary (FS):**
>   The model is *required* to output a `<summary>` after retrieval.
>   If the tag is missing, **reward = 0**.
>   This strict enforcement increases reward sparsity and significantly degrades performance (Appendix Table 5).
>
> - **Naive Summary (NS):**
>   The model *may optionally* produce a `<summary>` of retrieved documents.
>   A correct answer is rewarded even without a summary.
>
> - **Supportive-Evidence Notes (SEN, ours):**
>   Structured note-taking with explicit markings for key (`*`) and uncertain (`-`) information.
>   This fine-grained supervision leads to consistent improvements across all tasks.
>
> ---
>
>
> | Methods | TriviaQA F1 | TriviaQA EM | 2Wiki F1 | 2Wiki EM | Bamboogle F1 | Bamboogle EM | Musique F1 | Musique EM | PopQA F1 | PopQA EM | NQ F1 | NQ EM | HotpotQA F1 | HotpotQA EM |
> |--------|-------------|-------------|-----------|-----------|---------------|--------------|-------------|-------------|-----------|----------|--------|--------|--------------|--------------|
> | **Base** | 0.754 | 0.694 | 0.280 | 0.244 | 0.377 | 0.320 | 0.274 | 0.184 | 0.498 | 0.482 | 0.550 | 0.508 | 0.464 | 0.420 |
> | **+ FS** | 0.708 | 0.638 | 0.334 | 0.304 | 0.338 | 0.224 | 0.274 | 0.184 | 0.472 | 0.454 | 0.505 | 0.450 | 0.423 | 0.362 |
> | **+ NS** | 0.774 | 0.712 | 0.462 | 0.424 | **0.496** | **0.384** | 0.280 | 0.204 | 0.500 | 0.488 | 0.551 | 0.502 | **0.560** | **0.498** |
> | **+ SEN** | **0.795** | **0.730** | **0.505** | **0.464** | 0.440 | 0.352 | **0.317** | **0.210** | **0.524** | **0.514** | **0.563** | **0.518** | 0.550 | 0.482 |
>
> The comparison **SEN vs. NS** demonstrates that incorporating structured notes itself brings substantial improvements, even before considering reward bonuses.
>
> ---
>
> In summary:
> - If the answer is correct, then **reward = 1.0**
> - If the answer is correct **and an SEN is provided**, then **reward > 1.0 (extra bonus)**
>
> This design:
>
> - encourages note-taking without forcing it,
> - avoids reward sparsity,
> - prevents the degradation observed in FS, and
> - maintains stable, efficient training.
>
> We provide full experimental evidence in the **appendix following the main text (Table 5)**.
>
> We hope these clarifications assist in your evaluation, and we sincerely appreciate the AC, PC, and reviewers for your time and consideration toward acceptance.

---

### Author Response · Authors · 2025-12-02
**General Response**

We thank all reviewers and sincerely appreciate the AC and PC for taking the time to read our detailed clarifications and consider our submission.
Our work achieves 7% lower inference time and 20%–90% relative accuracy gains (Q3.1) over strong baselines, shows statistically significant performance advantages (Q4.4), and includes complete reproducibility materials.
vZFy noted our **"well-motivated problem"** and **"innovative strategy"**,vUdu highlighted our handling of **"noise and error accumulation"**. aKhP and 2RY5 praised the **"extensive experiments,"** confirming **"consistent performance gains"** (vZFy) and strong **"generalization"** (aKhP).


To fully address concerns **regarding reproducibility and experimental validity**, we have provided the following supplementary materials:

  -   **Full Codebase:** We provide the complete implementation for both training and testing pipelines.
  -   **Dataset:** The complete test set is included. For the training set, **we used the same standard public dataset as the baseline Search-R1**, which is publicly available.
  - **Extensive Ablation Results**: To guide future research, we present detailed design attempts in the appendix following the main text. We find:
    - Enforcing rigid behaviors noticeably harms model performance.
    - Evidence-oriented notes outperform generic summaries.
    - Documented negative design attempts to inform future work.

**Reviewer aKhP (Implementation & Advantage)** questioned the implementation and benefit of our Entailment Judge. **We include the full implementation in the supplementary materials**. Our lightweight judge achieves a **~1000× speedup and 48× size reduction** over LLM judges, enabling scalable process-based RL. Ablation studies further validate the feasibility of our reward design, aiming to address the reviewer’s concerns.


**Reviewer 2RY5 (Reproducibility & Efficiency)** raised concerns regarding code availability and efficiency. We have **released the full codebase** (including all prompts) to ensure complete reproducibility. In addition, our detailed analysis shows that the concise note-taking mechanism substantially reduces token overhead, yielding a **25%–39% reduction in output length** compared to the Search-R1 baseline.


**Reviewer vUdu (Novelty & Fairness)** expressed concerns that our experimental setting might differ from prior methods and suggested that our approach may not be fundamentally distinct. We clarify that **our training set is exactly the same as the baseline Search-R1 training set**, and we have also uploaded the full test sets to facilitate transparent verification. Furthermore, we additionally evaluated our method on an expanded and re-sampled test set, and the model **continues to show strong advantages**, confirming the robustness of our findings. All models are evaluated under **identical conditions** to ensure absolute fairness.
We further explain in the detailed responses that our workflow-oriented reward design is fundamentally different from prior approaches. Under these strictly controlled settings, this design yields substantial performance gains (e.g., **+91% on 2Wiki**, **+40% on Bamboogle**) while also reducing **Total Step (s)** by **15.4%**. These improvements, previously unattainable for prior methods, demonstrate the fundamental difference of our approach. **We also include extensive failed design attempts in the appendix at the end of the main PDF to help inform future model development.**


**Reviewer vZFy (Generalization & Efficiency)** raised concerns regarding generalization and the computational cost of the entailment judge.
To address the generalization concern, we conducted preliminary tests on a task-oriented dialogue setting. The results show that **SEN+EQR generalizes beyond QA**, improving task success from **17% → 21%** while also shortening conversations. This provides further evidence of the broader applicability of our method.
We provided comprehensive analyses showing that the judge adds only a **small, training-only overhead** (~4 ms per sample; ~17 s per step for batch=4096), while SEN+EQR provides clear efficiency gains: **−39.1% update time**, **−7% inference latency**, and a **−15.4% reduction in total step time**, along with a **35.2% lower update-time per token**. These results confirm that workflow-level supervision yields a more efficient and stable training and inference pipeline.
Multi-run evaluations further confirm **statistically significant improvements**, with our method achieving the **highest mean EM (0.443)** and tighter **95% CIs** than Search-R1, ensuring the gains are robust and not due to randomness.

Again, we sincerely appreciate the reviewers', AC's and PC’s valuable time and careful evaluation, and we hope that our clarifications, along with the paper’s **substantial performance and efficiency improvements**, help you assess whether we have fully addressed the reviewers’ concerns and make a fair, well-informed decision.

---

### Meta-Review · Area_Chair_PSbN · 2026-01-02

**Summary:**

List of main concerns:

1. Important details were missing: reward computation details, the prompt used to elicit notes from the model was not specified, training data details, whether SFT was used, how specific hyperparameter values were chosen. (aKhP, 2RY5, vZFy)
2. Missing evidence of efficiency improvements. (aKhP, vZFy)
3. Unclear whether the approach generalizes beyond QA. (vZFy)
4. Missing intrinsic evaluation of the accuracy of SENs. (vZFy)
5. Test sets are small and empirical setup not aligned with prior work. (vUdu)
6. The paper is missing error analysis. (vZFy)

Reviewers vUdu and vZFy were also concerned that the work is incremental, and that there exists prior work that explored similar ideas, although they have not cited specific relevant papers. So I have not given this concern much weight.

**Reviewer Concerns:**

Addressed concerns: I think concerns 1, 2 and 5 were fully addressed by the rebuttal. The authors provided additional details, results quantifying the effect of hyperparameter values, and detailed results that quantify efficiency improvements, and concern 5 was likely only a misunderstanding.

Partially addressed concerns:

- Concern 3: The authors presented results on a dialog task, which is different from the original QA tasks as the "reasoning unit is a dialogue turn rather than a retrieval hop" (quoted from the authors' response). It is not clear to me how this task is related to the main motivation behind the proposed method, which is to deal with a low signal-to-noise ratio and the distraction due to irrelevant information. One would expect those issues to be common when retrieval is involved, but I do not see how dialog modeling is a relevant task.

Not addressed:

- Concern 4: The authors argue that evaluating the accuracy of SENs would be computationally expensive when done within the training loop. However, that should not be relevant for offline evaluation to understand the impact of SENs on the final outputs. That said, given that SENs are shown to improve model performance on the end task, this may not be a very serious concern.
- Concern 6: The authors only show one specific kind of error in their response, and do not quantify how common it is.

**Reviewer Scores:**

aKhP: All concerns addressed, so likely increase to 6.
2RY5: Concern regarding missing analysis of when the model produces SENs was not fully addressed, so likely no increase.
vUdu: Concern regarding novelty was not addressed, but some misunderstanding resolved. So likely increase to 4.
vZFy: Concern regarding novelty not addressed, those regarding generalization beyond QA, missing error analysis and intrinsic evaluation not addressed. So likely no increase.

---

### Decision · Program_Chairs · 2026-01-26

Reject